# Neddylation regulates the development and function of glutamatergic neurons
Josefa Torres [1], Zehra Vural[1], Maksims Fiosins[2], Valentin Schwarze[1], Inés Hojas-García-Plaza[1], Fritz Benseler [1], Stefan Bonn [2], Silvio O. Rizzoli [3], Benjamin H. Cooper[1], JeongSeop Rhee[1], Nils Brose [1] & Marilyn Tirard[1] ✉

Neuronal development and function are orchestrated by a plethora of regulatory mechanisms that control the abundance, localization, interactions, and function of proteins. A key role in this regard is assumed by post-translational protein modifications (PTMs). While some PTM types, such as phosphorylation or ubiquitination, have been explored comprehensively, PTMs involving ubiquitin-like modifiers (Ubls) have remained comparably enigmatic (Ubls). This is particularly true for the Ubl Nedd8 and its conjugation to proteins, i.e. neddylation, in nerve cells. In the present study, we generated a conditional Nedd8 knock-out mouse line and examined the consequences of Nedd8-deletion in cultured post-mitotic glutamatergic neurons. Our findings reveal that Nedd8-ablation in young glutamatergic neurons causes alterations in the expression of developmental transcription factors that control neuronal differentiation, ultimately leading to defects in the development of a mature glutamatergic neuronal phenotype. Apparent manifestations of these defects include increased vGlut2 expression levels, reduced vGlut1 and endophilin1 expression levels, reduced dendrite complexity, and increased transmitter release probability. Collectively, our results highlight a pivotal role for neddylation in controlling the fate of glutamatergic neurons and excitatory synaptic transmission.

Neurons are post-mitotic cells that usually survive for the lifespan of their host organism. To allow for such long lifetimes, neuronal protein homeostasis is dynamically regulated and tightly controlled to support optimal functional efficacy and adaptability to changes in their environment. Synapses between neurons represent the prime communication units in the brain. They determine the spatiotemporal organization of neuronal inputs and outputs, thereby defining the layout and function of neuronal circuits, and their plasticity is a key determinant of memory processes. Accordingly, as with neurons in general, synaptic protein homeostasis and function are under tight dynamic control[1–3].

Post-translational protein modifications (PTM) are essential regulatory mechanisms in this regard, beyond processes that control gene transcription and mRNA translation. They expand the proteome by diversifying protein structure and function. Moreover, PTMs are at the core of many signaling pathways that enable cells to develop, differentiate, and rapidly respond or adapt to changes in their physiological status and environment[4–8].

The spectrum of known PTMs is broad, ranging from proteolytic processing and chemical modification of amino acid side chains to protein conjugation. Likewise, the functional consequences of PTMs are very diverse, including, among other aspects, alterations of protein interactions, activities, localization, or half-life[4,8,9]. As concerns neuronal development and function, an extensive number of studies demonstrated fundamental roles of multiple PTMs, such as phosphorylation and ubiquitination, in neuronal physiology and pathophysiology[9,10]. Indeed, the PTM of key transcription factors orchestrates the acquisition of cell type identity, neuronal differentiation, and synapse diversity[11–14], and the PTM of synaptic proteins regulates every step of the synapse fate, from synaptogenesis, via synapse maturation and plasticity, to synapse maintenance and elimination[6,8,15–17].

PTMs involving ubiquitin-like modifiers (Ubls, e.g., SUMO) have emerged as a novel protein regulatory principle of fundamental importance in many cell biological processes, operating at a level of complexity akin to phosphorylation or ubiquitination[4,8,10]. For instance, we and others showed that the sumoylation of multiple transcriptional regulators affects neuronal development and function, supporting the important notion that the PTM of key transcriptional regulators is essential for the determination of cell type identity, neuronal differentiation, and synapse diversity[18–23].

[1]Department of Molecular Neurobiology, Max Planck Institute for Multidisciplinary Sciences, Göttingen, Germany. [2]Institute of Medical Systems Biology, Center for Biomedical AI (bAIome), Center for Molecular Neurobiology (ZMNH), University Medical Center Hamburg-Eppendorf, Hamburg, Germany. [3]Department for Neuro- and Sensory Physiology, University Medical Center Göttingen, Göttingen, Germany. ✉e-mail: tirard@mpinat.mpg.de

Beyond SUMOs, Nedd8 (neuronal precursor cell expressed developmentally down-regulated 8) is arguably the second best characterized Ubl[24]. Like all other Ubls, Nedd8 is conjugated to lysine residues in target proteins via a three-step enzymatic cascade[4,25], involving an E1 (Nae1, heterodimer of Appb1 and Uba3), an E2 (Ubc12), and several E3 enzymes (RBX1/2, DCN1, MDM2, Rnf111), whereas several peptidases (Nedp1/Senp8, Uchl3, Csn5) ensure dynamics of this PTM[4]. Nedd8 knock-out mice, as well as mice lacking Nae1 or Ubc12, die during embryogenesis, establishing neddylation as an essential PTM[4].

As with other Ubls, neddylation has mainly been studied in highly proliferating cells, where it regulates a plethora of signaling events involved in the maintenance of DNA integrity or proteostasis[26,27]. In contrast, its role in post-mitotic cells, such as neurons, remains enigmatic. Several functional perturbation studies linked neddylation to neurite outgrowth, spine growth, synapse density, and synaptic transmission[28–33]. However, the Nedd8 targets at the basis of these processes are unknown. The most abundant Nedd8 targets in all cell types are the cullins, components of the large multi-protein cullin-RING E3 ubiquitin ligases[34,35] that regulate a myriad of cellular processes, including the development and dendrite arborization of neurons[36–38]. So far, only very few neuronal non-cullin Nedd8 targets have been described[39], including Cofilin, PSD95, and mGlu7[28,29,32], whose neddylation is thought to partially account for the morphological and functional defects seen in studies on the perturbation of neddylation[28–33].

However, a definitive and systematic understanding of the role of neddylation in neuronal development and synaptic function is lacking, not least because corresponding earlier approaches to perturb neddylation were either of unknown specificity (i.e., pharmacological inhibition of Nae1 with MLN4924), indirect (i.e., KO of Nedd8 peptidases), or targeting late neurodevelopmental stages, making data interpretation and generalizations difficult[28–30,33].

To obtain stringent and comprehensive insight into the role of neddylation in neurons, we generated a conditional Nedd8 mouse line and analyzed the impact of Nedd8 loss on neuronal development and synaptic transmission. We systematically assessed the morphology and function of Nedd8-deficient neurons in autaptic culture, and observed a key role for neddylation in controlling the fate of glutamatergic neurons and excitatory synaptic transmission.

## Results

Using CRISPR/Cas9, we generated a conditional Nedd8 knock-out (Nedd8cKO) mouse line by inserting LoxP sites before the second and after the third *Nedd8* exons (Fig. 1A). Western blot analysis of lysates from primary hippocampal neurons infected with a CRE-expressing virus confirmed the loss of Nedd8, as compared to neuronal cultures expressing red fluorescence protein (RFP) as a control (Fig. 1B, C). For all subsequent experiments, primary hippocampal neurons obtained from P0 Nedd8cKO pups were infected after one day in vitro (DIV1) with CRE-expressing virus to ablate Nedd8 expression (Nedd8-KO), RFP-expressing virus or no virus as control (CTRL), and were analyzed at DIV9–12.

### Nedd8-KO neurons show morphological defects but no change in synapse number

Many studies described defects in neuronal development, morphology and synapse number upon neddylation blockade[28,32,33]. Thus, we analyzed these parameters in Nedd8-deficient neurons as compared to RFP or non-infected control neurons (Fig. 1D–M). Sholl analysis revealed that the complexity of the neuronal dendritic tree was slightly altered upon Nedd8-deficiency, with a minor, but significant, reduction in the number of crossing dendrites within 50–100 μM from the soma (Fig. 1D–F). However, the total number of crossing dendrites and the total dendrite length were not reduced (Fig. 1G, H), indicating a moderate, but significant, impact of Nedd8 depletion on neuronal morphology.

Next, we quantified the number of excitatory synapses via immunolabeling using Synpasin1 as a pre-synaptic marker and PSD95 to label excitatory post-synapses (Fig. 1I–M); MAP2 was used to visualize overall

neuronal processes. The number of pre- and post-synaptic puncta was similar in Nedd8-KO as compared to the controls, e.g., RFP or non-infected CTRL neurons (Fig. 1J, K), leading to a similar number of pre- and post-synaptic colocalized puncta (Fig. 1L, M). Taken together, our data indicate that loss of Nedd8 expression alters neuronal morphology only moderately, without any impact on the number of synapses.

### Altered synaptic transmission in Nedd8-KO neurons

Next, we examined how Nedd8 depletion affected glutamatergic neurotransmission in autaptic hippocampal neurons (Figs. 2 and 3). While Nedd8-deficient autaptic neurons displayed no change in evoked excitatory post-synaptic currents (EPSCs) as compared to RFP or non-virus infected neurons (CTRL, Fig. 2A), CRE-infected neurons showed a 30% decrease in the readily-releasable pool (RRP) of synaptic vesicles (SV) as assessed by hypertonic sucrose stimulation (Fig. 2B), accompanied with a significant increase in synaptic transmitter release probability ($P_{vr}$, Fig. 2C). Importantly, there was no change in spontaneous miniature EPSC (mEPSC) amplitude (Fig. 2D), while a non-significant trend towards a reduction in mEPSC frequency was observed (Fig. 2E). Importantly, currents induced by fast superfusion of neurons with glutamate or GABA were not different between Nedd8-KO and the CTRL and RFP controls (Fig. 2F, G), indicating no differences in the surface expression of functional glutamate receptors. Thus, Nedd8 depletion does not profoundly alter post-synaptic function in a profound fashion.

We next studied synaptic short-term plasticity by measuring the synaptic response to high-frequency stimulation (HFS) (Fig. 2H–N). We found that the EPSC amplitude of CRE-infected neurons depressed faster than that of RFP or non-infected CTRL neurons (Fig. 2H, I). In addition, and in accord with the decreased RRP size as assessed by hypertonic sucrose stimulation (Fig. 2B), the paired pulse ratio (Fig. 2J, K) and the RRP size ($RRP_{40Hz}$) as estimated by back extrapolation of the cumulative EPSC during 40 Hz stimulus trains were decreased in Nedd8-deficient neurons as compared to RFP or non-infected CTRL neurons (Fig. 2L). Finally, the refiling rate of the RRP – i.e., the slope of the back extrapolated cumulative EPSC curve of the 40 Hz trains – showed a trend towards reduction that was not significant (Fig. 2M). When we normalized the RRP refiling rate to the $RRP_{40Hz}$, no difference between the tested group were detectable (Fig. 2N). Thus, the more pronounced short-term depression observed in Nedd8-deficient neurons (Fig. 2H, I) is primarily attributable to a reduction in RRP size and increase $P_{vr}$, rather than to an aberrant activity-dependent RRP refiling process.

### Normal VGCC activity but reduced SV-VGCC coupling in Nedd8-KO neurons

Regarding the changes in short-term plasticity observed in Nedd8-deficient neurons, we next examined the possibility of alterations in the properties of voltage-gated $Ca^{2+}$ channels (VGCCs). Given the serious challenges associated with studying VGCCs in the small presynaptic terminals of cultured hippocampal neurons, we investigated the VGCC properties in the soma of proximal dendrites. Initially, we assessed the overall density and voltage-dependence of VGCCs by measuring the current-voltage relationships, and found that Nedd8-loss had no discernible effect (Fig. 3A). We expanded our analyses by employing VGCC subtype-specific blockers to determine whether the influence of different presynaptic VGCCs in the presynaptic terminal on EPSC amplitude, RRP size, and $P_{vr}$ is altered by Nedd8-loss, by administering ω-Agatoxin and nimodipine, which specifically inhibit P/Q-type and L-type VGCCs, respectively (Fig. 3B–I). We found no significant differences between the groups tested, indicating that the altered RRP and $P_{vr}$ in Nedd8-KO neurons cannot be attributed to modifications in VGCCs properties.

In light of the results outlined above, particularly the $P_{vr}$ increase without changes in VGCC function, we next explored the possibility that Nedd8-loss selectively affects the priming and fusion of SVs that are more readily available for synaptic transmitter release due to VGCC-coupling. We performed electrophysiological recordings in the presence of EGTA-AM, a

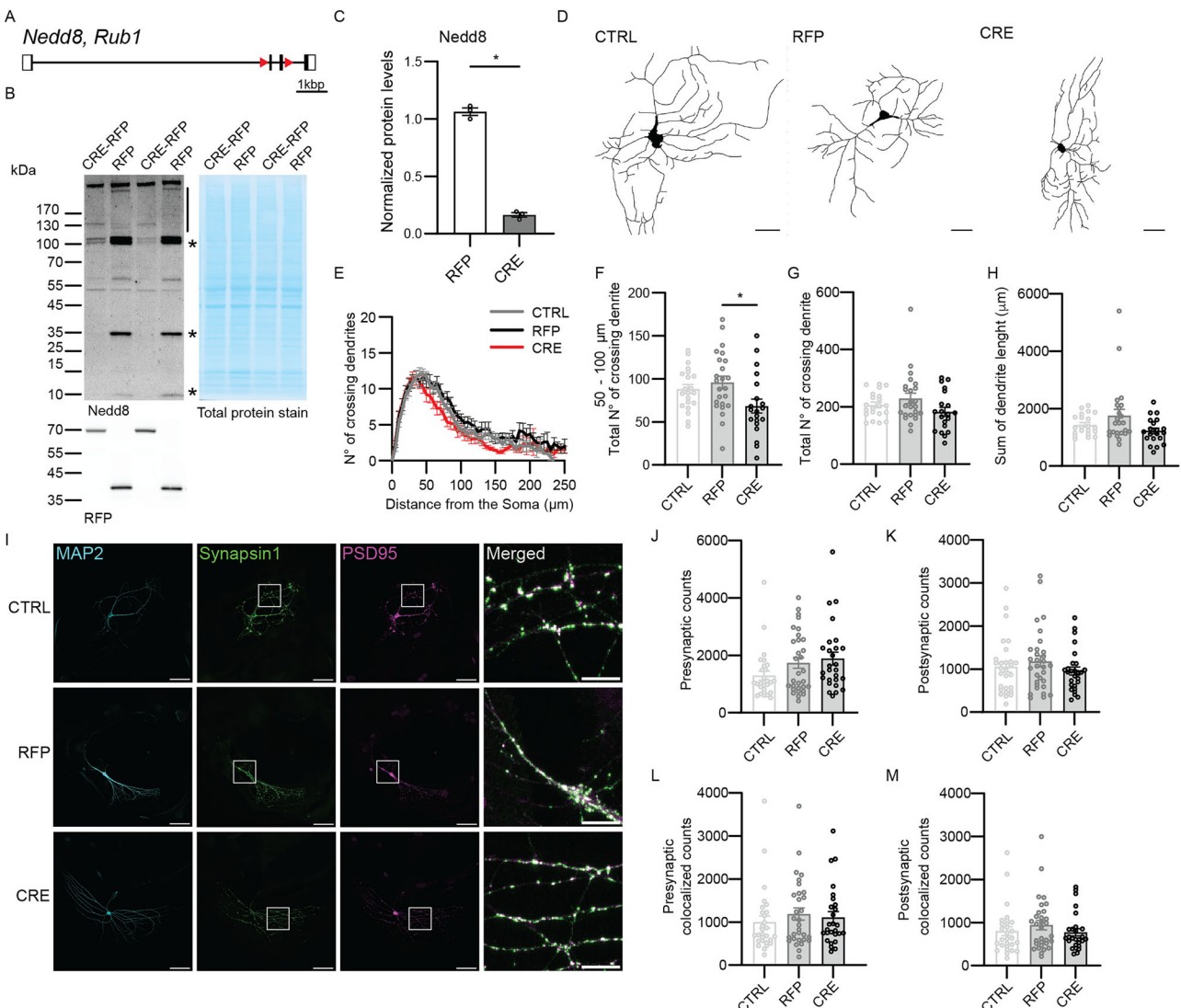

**Fig. 1 | Nedd8 deficiency alters dendritic branching but not synapse number.**
**A** On scale schematic view of the *Nedd8* mouse gene. LoxP sites (red triangles, not on scale) were inserted before the second exon and after the third exon using a CRISPR/Cas9 strategy. **B** Total protein stain (right) Anti-Nedd8 (top left) and RFP (bottom left) Western blot analysis of DIV12 primary hippocampal culture lysates induced with virus at DIV1 as indicated on top. The line and stars indicate Nedd8-specific signal. Molecular weight marker is indicated on the left (kDa). **C** Bar graph showing normalized Nedd8 protein levels calculated from Western blot analysis as depicted in (**B**). Nedd8 signal (black line and stars) was normalized to total protein stain ($N = 3$, $n_{RFP} = 3$, $n_{CRE} = 3$). Bars represent mean ± SEM. Data were compared using a one-tailed Mann-Whitney comparison test, where $P = 0.05$. **D** DIV13 Nedd8cKO autaptic hippocampal neurons infected at DIV1 with lentivirus expressing RFP or CRE-recombinase-RFP (CRE) were fixed and immunostained for MAP2. Neurons without lentiviral infection were used as infection controls (CTRL). Images were acquired using a confocal microscope, and representative binary images of MAP2-immunostained neurons are shown. Scale bar = 20 μm. **E** Line plots showing the number of dendritic intersections as a function of distance from the soma obtained by Sholl analysis ($N = 3$, $n_{CTRL} = 22$, $n_{RFP} = 26$, $n_{CRE} = 20$ cells). **F** Bar graph showing the number of crossing dendrites between 50 and 100 μm from the soma, calculated from Sholl analysis ($N = 3$; $n_{CTRL} = 22$, $n_{RFP} = 26$, $n_{CRE} = 20$ cells). Data was compared using Tukey's multiple comparisons test, where $P_{CTRL-RFP} = 0.72$, $P_{CTRL-CRE} = 0.14$, $P_{RFP-CRE} = 0.020$. **G** Bar graphs showing the total number of crossing dendrites, calculated from Sholl analysis ($N = 3$; $n_{CTRL} = 22$, $n_{RFP} = 26$, $n_{CRE} = 20$ cells). Data was compared using Dunn's multiple

comparisons test, where $P_{CTRL-RFP} > 0.99$, $P_{CTRL-CRE} = 0.53$, $P_{RFP-CRE} = 0.21$. **H** Bar graph showing the dendrite length sum ($N = 3$; $n_{CTRL} = 22$, $n_{RFP} = 26$, $n_{CRE} = 20$ cells). Data was compared using Dunn's multiple comparisons test, where $P_{CTRL-RFP} > 0.99$, $P_{CTRL-CRE} = 0.35$, $P_{RFP-CRE} = 0.11$. **I** Immunostaining of autaptic hippocampal neurons with antibodies against MAP2 (cyan), synapsin1 (green), and PSD95 (magenta). Scale bar = 50 μm. The white square indicates a closer view depicted in the right column of panels. Scale bar = 10 μm. **J** Bar graph showing the total number of synapsin1 (presynaptic) counts. ($N = 3$; $n_{CTRL} = 28$, $n_{RFP} = 31$, $n_{CRE} = 27$ cells). Data was compared using Dunn's multiple comparisons test, where $P_{CTRL-RFP} = 0.37$, $P_{CTRL-CRE} = 0.063$, $P_{RFP-CRE} > 0.99$. **K** Bar graph showing the total number of PSD95 (postsynaptic) counts ($N = 3$; $n_{CTRL} = 28$, $n_{RFP} = 31$, $n_{CRE} = 27$ cells). Data was compared using Dunn's multiple comparisons test, where $P_{CTRL-RFP} > 0.99$, $P_{CTRL-CRE} > 0.99$, $P_{RFP-CRE} = 0.65$. **L** Bar graph showing the total number of synapsin1 counts colocalized with PSD95 signal ($N = 3$; $n_{CTRL} = 28$, $n_{RFP} = 31$, $n_{CRE} = 27$ cells). Data was compared using Dunn's multiple comparisons test, where $P_{CTRL-RFP} = 0.73$, $P_{CTRL-CRE} = 0.94$, $P_{RFP-CRE} > 0.99$. **M** Bar graphs showing the total number of PSD95 counts colocalized with synapsin1 ($N = 3$; $n_{CTRL} = 28$, $n_{RFP} = 31$, $n_{CRE} = 27$ cells). Data was compared using Dunn's multiple comparisons test, where $P_{CTRL-RFP} = 0.77$, $P_{CTRL-CRE} > 0.99$, $P_{RFP-CRE} > 0.99$. In graph (**J–M**), dots correspond to the total number of synaptic counts obtained from one individual neuron. For all graphs, error bars represent mean ± SEM. Data were compared using a Mann-Whitney, Kruskal-Wallis, Dunn´s multiple comparison test, and a Tukey's multiple comparisons test, where * $p < 0.05$.

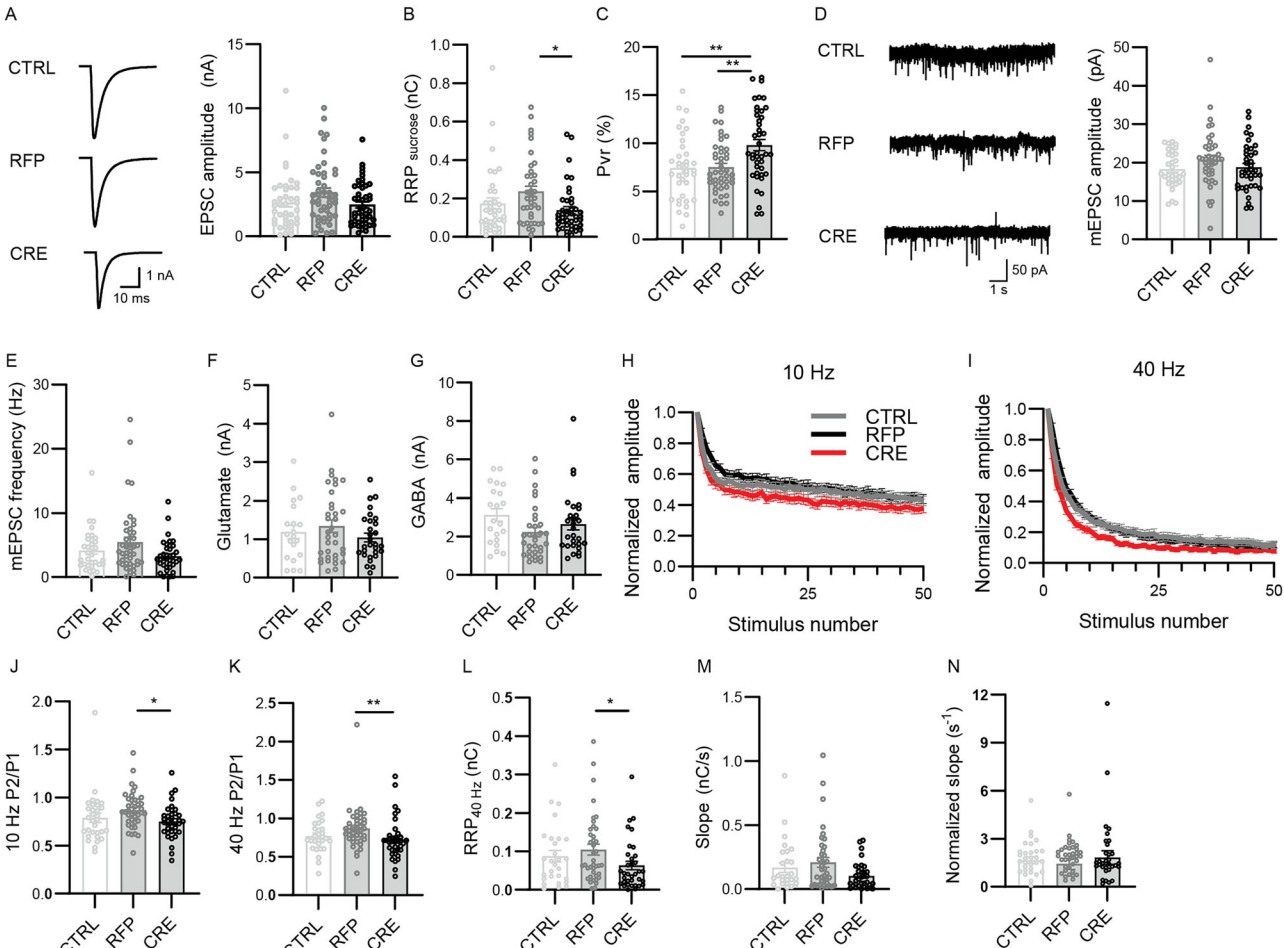

**Fig. 2 | Nedd8 knock-down alters synaptic release probability.** Autaptic excitatory hippocampal neurons from Nedd8cKO animals were infected at DIV1 with lentivirus expressing RFP or CRE-recombinase-RFP (CRE) and analyzed at DIV11-13. Neurons without lentiviral infection were used as infection controls (CTRL).
**A** Representative traces (left) and bar graph (right) of evoked EPSC amplitudes of autaptic neurons infected as indicated below the graphs ($N = 5$; $n_{CTRL} = 41$, $n_{RFP} = 47$, $n_{CRE} = 47$ cells). Data was compared using Dunn's multiple comparisons test, where $P_{CTRL-RFP} = 0.22$, $P_{CTRL-CRE} > 0.99$, $P_{RFP-CRE} > 0.18$. **B** Bar graph depicting the total charge transferred by the release of the RRP obtained after treatment with hypertonic sucrose solution ($N = 5$; $n_{CTRL} = 36$, $n_{RFP} = 44$, $n_{CRE} = 42$ cells). Data was compared using a Dunn's multiple comparisons test, where $P_{CTRL-RFP} = 0.10$, $P_{CTRL-CRE} > 0.99$, $P_{RFP-CRE} = 0.011$. **C** Bar graph showing the probability of vesicular release ($P_{vr}$) of individual neurons ($N = 5$; $n_{CTRL} = 36$, $n_{RFP} = 44$, $n_{CRE} = 42$ cells). $P_{vr}$ is calculated by dividing the charge transfer during an evoked EPSC by the charge transfer during the sucrose response, and then expressed as a percentage. Data was compared using a Tukey's multiple comparisons test, where $P_{CTRL-RFP} = 0.99$, $P_{CTRL-CRE} = 0.0055$, $P_{RFP-CRE} = 0.0047$. **D** Representative traces (left) and bar graph (right) depicting spontaneous mEPSC amplitudes of autaptic neurons treated with 300 nM TTX ($N = 5$; $n_{CTRL} = 33$, $n_{RFP} = 43$, $n_{CRE} = 39$ cells). Data was compared using a Dunn's multiple comparisons test, where $P_{CTRL-RFP} = 0.33$, $P_{CTRL-CRE} > 0.99$, $P_{RFP-CRE} = 0.34$. **E** Bar graph showing the frequency of mEPSC measured in autaptic neurons in the presence of 300 nM TTX ($N = 5$; $n_{CTRL} = 33$, $n_{RFP} = 43$, $n_{CRE} = 39$ cells). Data was compared using a Dunn's multiple comparisons test, where $P_{CTRL-RFP} = 0.99$, $P_{CTRL-CRE} = 0.86$, $P_{RFP-CRE} = 0.091$. **F** Bar graph showing the amplitude of the peak current generated by the superfusion of 100 μM Glutamate ($N = 5$; $n_{CTRL} = 20$, $n_{RFP} = 34$, $n_{CRE} = 29$ cells). Data was compared using a Dunn's multiple comparisons test, where $P_{CTRL-RFP} > 0.99$, $P_{CTRL-CRE} > 0.99$, $P_{RFP-CRE} > 0.99$. **G** Bar graph showing the amplitude of the peak current generated by the superfusion of 3 μM GABA ($N = 5$; $n_{CTRL} = 21$, $n_{RFP} = 32$,

$n_{CRE} = 28$ cells). Data was compared using a Dunn's multiple comparisons test, where $P_{CTRL-RFP} = 0.053$, $P_{CTRL-CRE} = 0.62$, $P_{RFP-CRE} = 0.72$. **H** Line plots showing the change in EPSC amplitudes during a 10 Hz stimulation train ($N = 5$; $n_{CTRL} = 31$, $n_{RFP} = 41$, $n_{CRE} = 37$ cells). Data were normalized to the first response of the respective train and shown as mean $+/-$ SEM. **I** Line plots showing the change in EPSC amplitudes during a 40 Hz stimulation train ($N = 5$; $n_{CTRL} = 29$, $n_{RFP} = 42$, $n_{CRE} = 37$ cells). Data were normalized to the first response of the respective train and shown as mean $+/-$ SEM. **J** Bar graph showing paired-pulse ratios obtained from 10 Hz stimulation trains ($N = 5$; $n_{CTRL} = 31$, $n_{RFP} = 41$, $n_{CRE} = 37$ cells). Data was compared using a Dunn's multiple comparisons test, where $P_{CTRL-RFP} = 0.088$, $P_{CTRL-CRE} > 0.99$, $P_{RFP-CRE} = 0.013$. **K** Bar graph showing paired-pulse ratios obtained from 40 Hz stimulation trains ($N = 5$; $n_{CTRL} = 29$, $n_{RFP} = 42$, $n_{CRE} = 37$ cells). Data was compared using a Dunn's multiple comparisons test, where $P_{CTRL-RFP} = 0.23$, $P_{CTRL-CRE} = 0.47$, $P_{RFP-CRE} = 0.0017$. **L** Bar graphs showing the RRP size as estimated by back extrapolation of the cumulative EPSC after 40 Hz stimulus trains ($N = 5$; $n_{CTRL} = 28$, $n_{RFP} = 40$, $n_{CRE} = 35$ cells). Data was compared using a Dunn's multiple comparisons test, where $P_{CTRL-RFP} > 0.99$, $P_{CTRL-CRE} = 0.48$, $P_{RFP-CRE} = 0.039$. **M** Bar graphs showing the slope of the back extrapolated cumulative EPSC curve of the 40 Hz trains ($N = 5$; $n_{CTRL} = 28$, $n_{RFP} = 39$, $n_{CRE} = 35$ cells). Data was compared using a Dunn's multiple comparisons test, where $P_{CTRL-RFP} > 0.99$, $P_{CTRL-CRE} = 0.65$, $P_{RFP-CRE} = 0.12$. **N** Bar graph showing the ratio between the slope of the back extrapolated cumulative EPSC curve (shown in **M**), divided by the $RRP_{40Hz}$ (shown in **L**), ($N = 5$, $n_{ctrl} = 32$, $n_{rfp} = 39$, $n_{cre} = 34$ cells). Data was compared using a Dunn's multiple comparisons test, where $P_{CTRL-RFP} > 0.99$, $P_{CTRL-CRE} > 0.99$, $P_{RFP-CRE} > 0.99$. In each graph, dots correspond to measurements recorded from individual neurons and bars represent mean ± SEM. Data were compared using a Kruskal-Wallis, Tukey's, Dunn's multiple comparison test, where $*p < 0.05$, $**p < 0.01$.

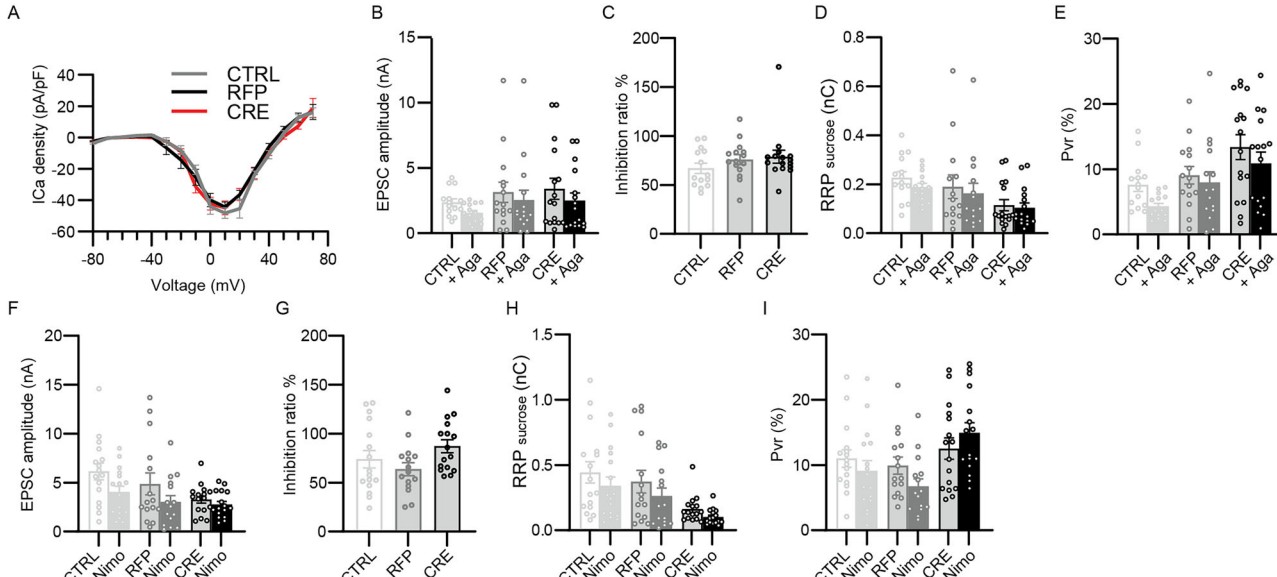

**Fig. 3 | VGCC currents are unchanged in Nedd8-KO neurons.** Analysis of calcium currents obtained from DIV9-13 Nedd8cKO autaptic excitatory hippocampal neurons infected at DIV1 with lentivirus expressing RFP, CRE-recombinase-RFP (CRE) or non-infected neurons (CTRL). **A** Current-voltage relationship of HVA calcium channels ($N = 3$; $n_{CTRL} = 23$, $n_{RFP} = 21$, $n_{CRE} = 23$ cells). **B** Bar graph showing the change in evoked EPSC amplitudes after treatment with 0.2 μM ω-Agatoxin (Aga) ($N = 2$; $n_{CTRL} = 14$, $n_{RFP} = 15$, $n_{CRE} = 16$ cells). Data was compared using a Dunn's multiple comparisons test, where $P_{CTRL-Aga} > 0.99$, $P_{RFP-Aga} > 0.99$, $P_{CRE-Aga} > 0.99$. **C** Bar graph showing the inhibition ratio of EPSC amplitudes after treatment with 0.2 μM ω-Agatoxin ($N = 2$; $n_{CTRL} = 14$, $n_{RFP} = 15$, $n_{CRE} = 16$ cells). Data was compared using a Dunn's multiple comparisons test, where $P_{CTRL-Aga} = 0.60$, $P_{RFP-Aga} = 0.90$, $P_{CRE-Aga} > 0.99$. **D** Bar graph depicting the total charge transferred by the release of the RRP upon application with hypertonic sucrose solution after treatment with 0.2 μM ω-Agatoxin (Aga) ($N = 2$; $n_{CTRL} = 14$, $n_{RFP} = 15$, $n_{CRE} = 16$ cells). Data was compared using a Dunn's multiple comparisons test, where $P_{CTRL-Aga} > 0.99$, $P_{RFP-Aga} > 0.99$, $P_{CRE-Aga} > 0.99$. **E** Bar graph showing the probability of vesicular release of individual neuron after treatment with 0.2 μM ω-Agatoxin (Aga) ($N = 2$; $n_{CTRL} = 14$, $n_{RFP} = 14$, $n_{CRE} = 16$ cells). Data was compared using a Dunn's multiple comparisons test, where $P_{CTRL-Aga} = 0.69$,

$P_{RFP-Aga} > 0.99$, $P_{CRE-Aga} > 0.99$. **F** Bar graph showing the change in EPSC amplitudes after treatment with 10 μM Nimodipine (Nimo) ($N = 2$; $n_{CTRL} = 16$, $n_{RFP} = 15$, $n_{CRE} = 16$ cells). Data was compared using a Tukey's multiple comparison test, where $P_{CTRL-Nimo} = 0.36$, $P_{RFP-Nimo} = 0.58$, $P_{CRE-Nimo} = 0.99$. **G** Bar graph showing the inhibition ratio of EPSC amplitudes after treatment with 10 μM Nimodipine ($N = 2$; $n_{CTRL} = 16$, $n_{RFP} = 15$, $n_{CRE} = 16$ cells). Data was compared using a Tukey's multiple comparisons tests, where $P_{CTRL-RFP} = 0.61$, $P_{CTRL-CRE} = 0.42$, $P_{RFP-CRE} = 0.083$. **H** Bar graph of the total charge transferred by the release of the RRP upon application with hypertonic sucrose solution after treatment with 10 μM Nimodipine (Nimo) ($N = 2$; $n_{CTRL} = 16$, $n_{RFP} = 15$, $n_{CRE} = 16$ cells). Data was compared using a Dunn's multiple comparisons test, where $P_{CTRL-Nimo} > 0.99$, $P_{RFP-Nimo} > 0.99$, $P_{CRE-Nimo} > 0.99$. **I** Bar graph showing the probability of vesicular release of individual neurons after treatment with 10 μM Nimodipine (Nimo) ($N = 2$; $n_{CTRL} = 16$, $n_{RFP} = 15$, $n_{CRE} = 16$ cells). Data was compared using a Tukey's multiple comparison test, where $P_{CTRL-Nimo} = 0.99$, $P_{RFP-Nimo} = 0.82$, $P_{CRE-Nimo} = 0.95$. In each graph, dots correspond to measurements recorded from individual neurons and bars represent mean ± SEM. Data were compared using an unpaired $t$ test, Mann-Whitney test, a Kruskal-Wallis and Dunn´s multiple comparison test.

slow $Ca^{2+}$ chelator used to perturb VGCC coupling to SV $Ca^{2+}$ sensor[40,41] (Fig. 4A). EGTA-AM reduced the $P_{vr}$ to a similar extent in Nedd8-KO neurons as compared to RFP infected neurons (Fig. 4B–D). Strikingly, however, EGTA-AM reversed synaptic depression during trains of action potentials in RFP-infected neurons (Fig. 4E, G) but not in CRE-expressing Nedd8-deficient cells (Fig. 4F, H). These data are compatible with the notion that a pool of ready-releasable SVs present near VGCCs is altered in Nedd8-deficient neurons (Fig. 4A).

## Normal ultrastructural organization of Nedd8-KO synapses at rest

We conducted a 3D ultrastructural analysis of synapses in cultured neurons to investigate possible ultrastructural correlates of the presynaptic functional deficits observed in Nedd8-KO neurons. We performed dual-axis TEM tomography to quantify the size and distribution of SV pools in presynaptic terminals (Supplementary Fig. 1). Tomographic analyses of high-pressure frozen and freeze-substituted primary hippocampal neurons infected with either RFP as a control or with CRE to deplete Nedd8 revealed no major morphological differences (Supplementary Fig. 1A–L). The cumulative frequency distribution (Supplementary Fig. 1E) and total number of SVs within 0–200 nm of the AZ (Supplementary Fig. 1F) were very similar between conditions. Likewise, an analysis of nearest-neighbor distances for all SVs within 0–200 nm of the AZ did not reveal any statistically significant differences in SV cluster characteristics between conditions (Supplementary

Fig. 1G). Morphological analyses were subsequently restricted to vesicles localized in close proximity to the AZ membrane (0–40 nm from the AZ, Supplementary Fig. 1H–J). No statistically significant differences were observed in either the spatial distribution of vesicles (Supplementary Fig. 1H), mean AZ areas (Supplementary Fig. 1H, Insert), density of docked vesicles per unit AZ (Supplementary Fig. 1I), or inter-vesicular nearest neighbor distances (Supplementary Fig. 1I, J). Finally, SV diameters were not distinguishable between Nedd8-deficient and RFP control neurons (Supplementary Fig. 1K, L). No difference in the number of morphologically docked SVs, which serve as a proxy for the number of molecularly primed SVs comprising the RRP[42], was observed between genotypes. Altogether, Nedd8-deficient synapses do not display any observable alteration in organization at AZ release sites at rest. This indicates that the functional defects described above manifest without incurring structural changes detectable by our methodology and originate from deeper and molecular Nedd8-dependent mechanisms.

## Altered expression of genes encoding key pre-synaptic proteins in Nedd8-KO neurons

In a first attempt to explore the molecular pathways by which Nedd8-loss causes the functional phenotypes we observed, we performed RNA-sequencing and differential expression analysis of CRE-expressing neurons vs. RFP-expressing neurons (Fig. 5A, Supplementary Fig. 2–4 and Supplementary Data 2). The underlying motivation was to not only determine

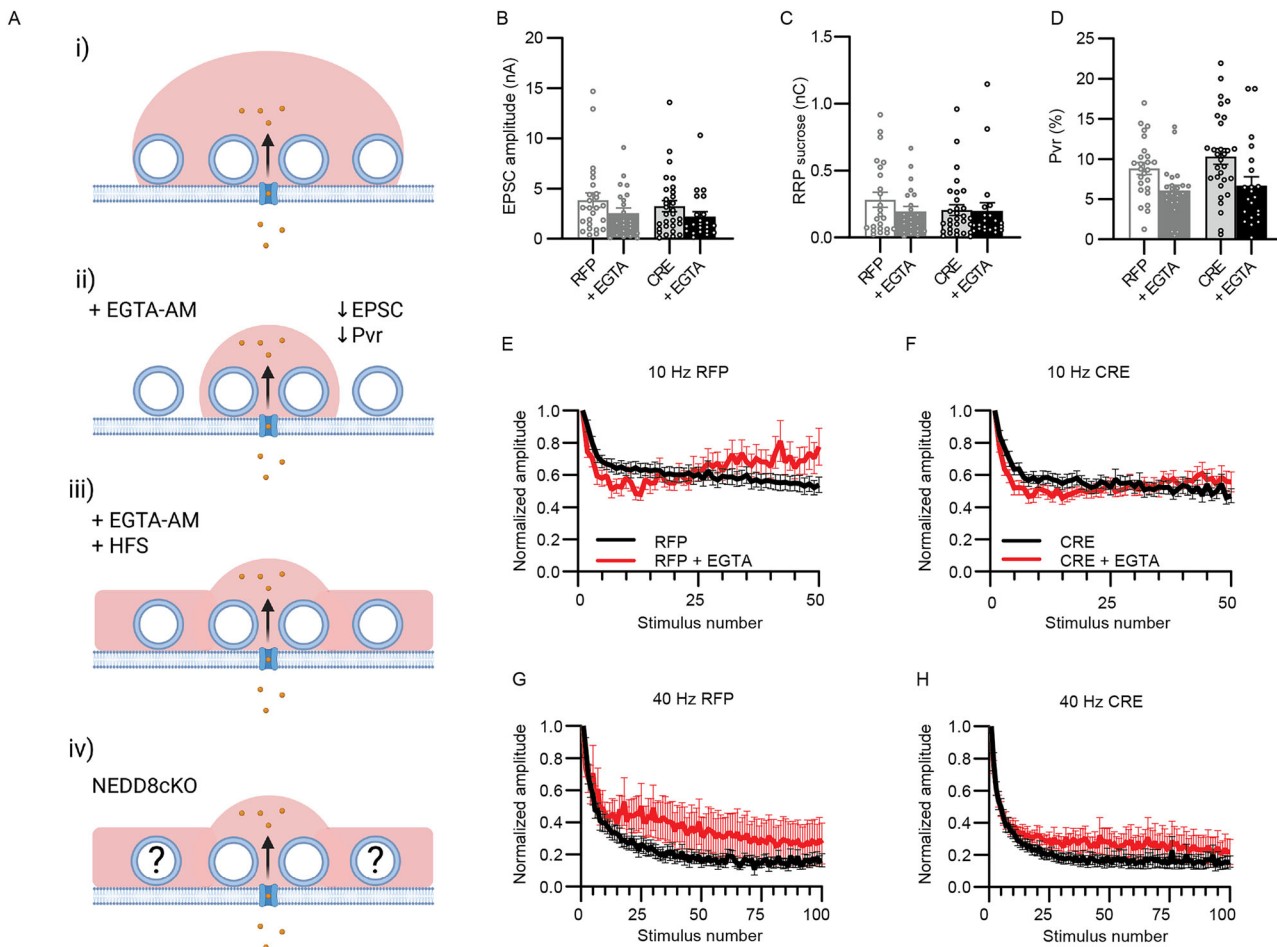

**Fig. 4 | Nedd8 regulates the number of synaptic vesicles coupled to VGCC.**
Recordings were performed using DIV9-13 Nedd8cKO autaptic excitatory hippocampal neurons infected at DIV1 with lentivirus expressing RFP, CRE-recombinase-RFP (CRE) or non-infected neurons (CTRL). **A** Schematic representation of the effect of EGTA-AM on synaptic vesicles coupled to calcium channels. The size of proteins and synaptic vesicles is not on scale. This image was generated with the help of Biorender. (i) Following an action potential, voltage-gated calcium channels (VGCC) open and $Ca^{2+}$ ions access the intracellular compartment. The red cloud indicates the area and number of vesicles impacted by the entry of calcium. (ii) After treatment with the permeable ester, EGTA-AM, hydrolyzation of intracellular esterases leads to the production of EGTA, which lowers the intracellular calcium concentration (reduction in the size of the red cloud), except in the vicinity of VGCC, leading to a decrease in the number of released vesicles. Ultimately, EPSC amplitude and $P_{vr}$ are reduced, but not the size of the RRP, as measured by hypertonic sucrose treatment. (iii) During a high-frequency train of action potentials (HFS), the effect of EGTA is restored due to the accumulation of calcium ions in the intracellular compartment. (iv) Working model: Nedd8-deficient neurons present fewer synaptic vesicles capable of being released. **B** Bar graph showing EPSC amplitudes, with and without treatment with 100 μM EGTA-AM ($N$ = 3; $n_{RFP}$ = 14, $n_{RFP+EGTA-AM}$ = 16, $n_{CRE}$ = 20, $n_{CRE+EGTA-AM}$ = 15 cells). Data was compared using a Mann-Whitney test, where $P_{RFP-}$

$_{CRE}$ = 0.53. **C** Bar graph of the total charge transferred by the release of the RRP upon application of hypertonic sucrose solution, with and without treatment with 100 μM EGTA-AM ($N$ = 3; $n_{RFP}$ = 14, $n_{RFP+EGTA-AM}$ = 17, $n_{CRE}$ = 19, $n_{CRE+EGTA-AM}$ = 15 cells). Data was compared using a Mann-Whitney test, where $P_{RFP-CRE}$ = 0.24. **D** Bar graph showing the impact of EGTA-AM treatment on the probability of vesicular release of individual neuron ($N$ = 3; $n_{RFP}$ = 14, $n_{RFP+EGTA-AM}$ = 17, $n_{CRE}$ = 20, $n_{CRE+EGTA-AM}$ = 15 cells). Data was compared using an unpaired $t$ test, where $P_{RFP-CRE}$ = 0.24. **E** Line plots showing the change in EPSC amplitudes during a 10 Hz stimulation train in neurons infected with RFP virus, with and without application of 100 μM EGTA-AM ($N$ = 3; $n_{RFP}$ = 15, $n_{RFP+EGTA-AM}$ = 16 cells). **F** Line plots showing the change in EPSC amplitudes during a 10 Hz stimulation train in neurons infected with CRE virus, with and without application of 100 μM EGTA-AM ($N$ = 3; $n_{CRE}$ = 19, $n_{CRE + EGTA-AM}$ = 14 cells). **G** Line plots showing the change in EPSC amplitudes during a 40 Hz stimulation train in neurons infected with RFP virus, with and without application of 100 μM EGTA-AM ($N$ = 3; $n_{RFP}$ = 15, $n_{RFP+EGTA-AM}$ = 16 cells). **H** Line plots showing the change in EPSC amplitudes during a 40 Hz stimulation train in neurons infected with CRE virus, with and without application of 100 μM EGTA-AM ($N$ = 3; $n_{CRE}$ = 19, $n_{CRE+EGTA-AM}$ = 14 cells). In each graph, dots correspond to measurements recorded from individual neurons and bars represent mean ± SEM. Data were compared using an unpaired $t$ test and a Mann-Whitney test.

direct transcriptional changes upon Nedd8-loss, but also to assess indirect compensatory mechanisms. Beyond a general analysis, we put a special focus on genes encoding synaptic proteins by using the SynGO database[43].

Upon stringent filtering, the expression of 566 genes remained significantly altered in Nedd8-deficient neurons (Fig. 5A; Log2FC of 1.3; -log10 $p$-value of 6), with 77% (436) up- and 23% (130) down-regulated. Among these were 66 SynGO genes that remained significantly changed after filtering. Interestingly, half of these were up- and the other half down-regulated (Fig. 5A), which strongly contrasts with the remodeling of the rest of the transcriptome induced by Nedd8-deficiency, indicating that Nedd8-

loss causes a general increase in gene expression but a more pronounced downregulation of genes expressing synaptic proteins.

SynGO pathway analysis showed that Nedd8-deletion leads to decreased expression of multiple mRNAs encoding pre-synaptic components, in particular proteins involved in the SV cycle (Supplementary Fig. 3) and to the up-regulation of genes related to the extracellular matrix or protein translation (Supplementary Fig. 4). Strikingly, vGlut1 (*Slc17a7*) was strongly down-regulated upon Nedd8-loss, while vGlut2 was up-regulated (*Slc17a6*, Fig. 5A–C). This result was further validated via RT-qPCR and at the protein level (Fig. 5D–H). Given the known correlation between $P_{vr}$ and

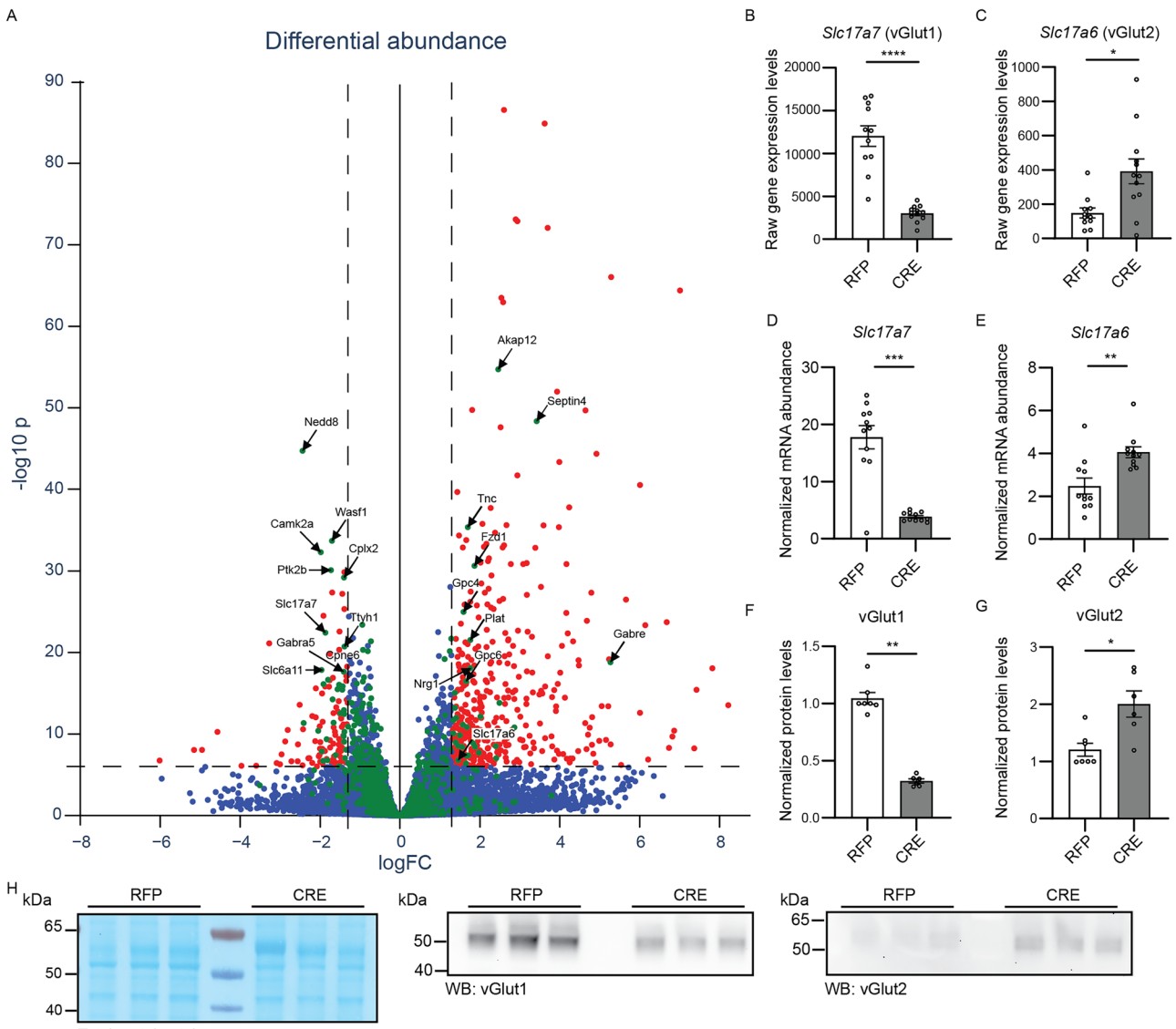

**Fig. 5 | Nedd8 depletion alters the expression of synaptic proteins. A** RNA was extracted from DIV13 Nedd8cKO primary hippocampal neurons infected at DIV1 with lentivirus expressing RFP or CRE-recombinase-RFP (CRE) and sequenced. Each dot represents a single gene (blue dots). Volcano plot depicts each gene based on their Log2 fold change (LogFC) and the reverse log10 of their $p$-value (-log10p). Red dots correspond to genes with a LogFC above 1,3 and -log10p above 6. Green dots correspond to genes found in SynGO. **B, C** Bar graphs showing the raw gene expression levels of *Slc17a7* (vGlut1, **B**) and *Slc17a6* (vGlut2, **C**) ($N = 3$, $n_{RFP} = 11$, $n_{CRE} = 12$). Data was compared using an unpaired $t$ test, where $P_{Slc17a7} < 0.0001$, and a Mann-Whitney test, where $P_{Slc17a6} = 0.011$. **D, E** Bar graphs showing the normalized mRNA expression levels of *Slc17a7* (vGlut1, **D**) and *Slc17a6* (vGlut2, **E**)

($N = 3$, $n_{RFP} = 11$, $n_{CRE} = 11$). Data was compared using a Mann-Whitney test, where $P_{Slc17a7} = 0.0006$, $P_{Slc17a6} = 0.0010$. **F, G** Bar graph depicting normalized protein levels of vGlut1 and vGlut2, as assessed by Western blot showed in (**H**). Data was compared using a Mann-Whitney test, where $P_{vGlut1} = 0.0012$, $P_{vGlut2} = 0.013$. **H** Total protein stain (left panel), and anti-vGlut1 (middle panel) and vGlut2 (right panel) Western blot analysis of primary hippocampal Nedd8cKO neurons lysates infected with lentivirus expressing RFP or CRE-recombinase-RFP (CRE). Molecular weight is indicated on the left side (kDa). For all bar graphs, data represent mean ± SEM. Data were compared using an unpaired $t$ test or a Mann-Whitney comparison test, where *$p < 0.05$, **$p < 0.01$, ***$p < 0.001$, ****$p < 0.0001$.

the resident isoform of vGlut expressed, with vGlut2-expressing neurons exhibiting higher $P_{vr}$[44,45], we further studied this phenomenon.

### Reduced density of vGlut1-containing synapses in Nedd8-KO neurons

We next compared vGlut1 immunoreactivity in Nedd8-deficient neurons compared to RFP-expressing controls (Fig. 6). Whereas the total number of synapses was unchanged (Fig 1I–M), the total number of vGlut1 puncta was significantly reduced in Nedd8-deficient neurons (Fig. 6A–C). Importantly, vGlut1 signal intensity was not altered in vGlut1-containing mutant synapses (Fig. 6D, E). STED imaging confirmed that the intensity of vGlut1 puncta along dendrites was unchanged and showed further that the sub-synaptic

distribution of vGlut1 is similar between Nedd8-deficient and control neurons (Fig. 6F–H). Our data indicate that the number of synapses containing robust levels of vGlut1 is decreased in Nedd8-KO neurons, whereas the synaptic enrichment of vGlut1 is unchanged in synapses that still contain vGlut1.

### $P_{vr}$ defects in Nedd8-KO neurons are not reverted by vGlut2 knock-down or vGlut1 overexpression

Our data show that vGlut2 expression is up-regulated by 50% and vGlut1 down-regulated by 60–70% upon Nedd8-loss (Fig. 5). This finding could indicate a delayed differentiation of glutamatergic neurons, which switch from vGlut2 to vGlut1 expression as they mature, or a complex scenario of vGlut1 depletion compensated by vGlut2 expression[46]. Previous studies

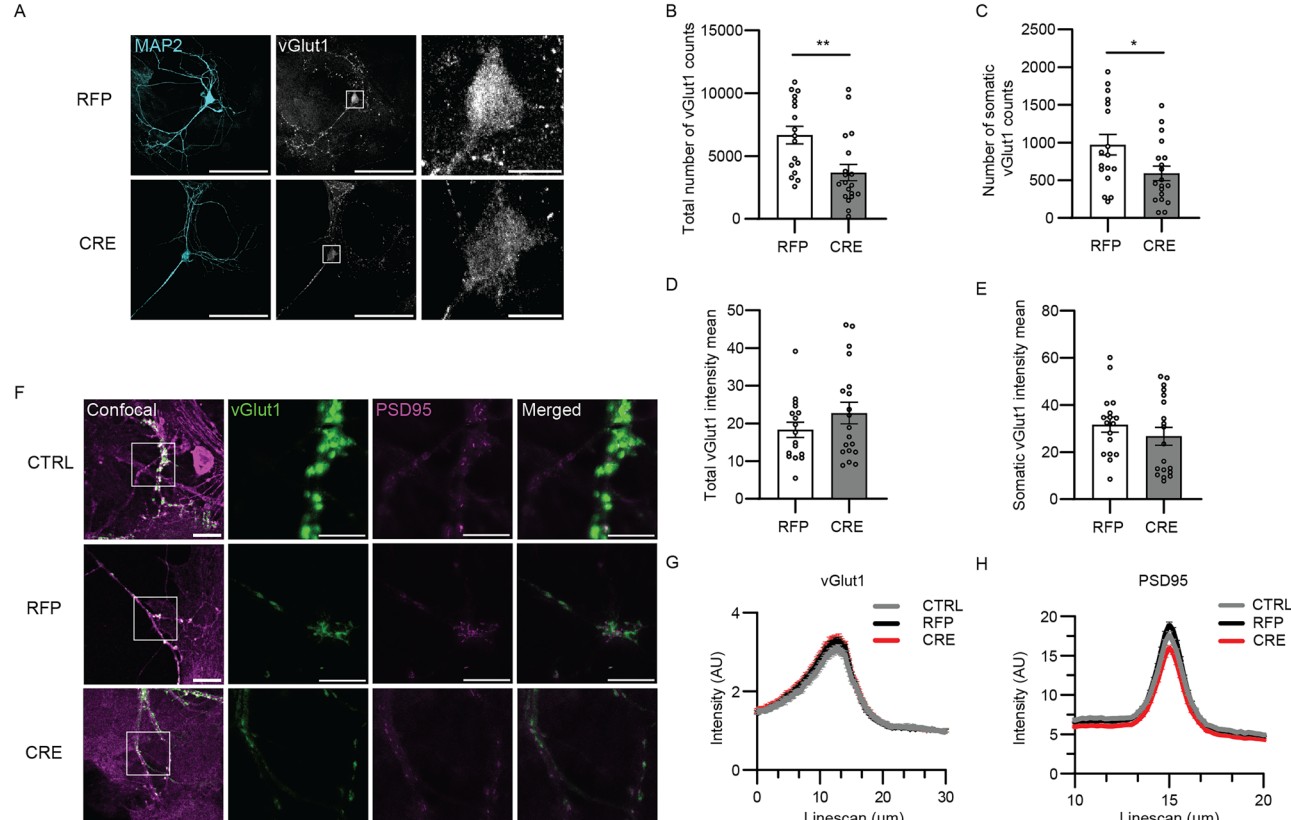

**Fig. 6 | vGlut1 levels are altered upon Nedd8 depletion.** DIV13 Nedd8cKO autaptic hippocampal neurons infected at DIV1 with lentivirus expressing RFP or CRE-recombinase-RFP (CRE) were fixed and immunostained with antibodies as indicated. Neurons without lentiviral infection were used as infection controls (CTRL). **A** Representative confocal images of autaptic neurons stained with antibodies against MAP2 and vGlut1. Scale bars = 80 (top panels) and 10 μm (bottom panels). **B** Bar graph showing the total number of vGlut1 counts ($N = 2$; $n_{RFP} = 17$, $n_{CRE} = 19$ cells). Data was compared using a Mann-Whitney test, where $P = 0.0020$. **C** Bar graph showing the number of somatic vGlut1 counts ($N = 2$; $n_{RFP} = 17$, $n_{CRE} = 19$ cells). Data was compared using an unpaired $t$ test, where $P = 0.026$. **D** Bar graph showing the total vGlut1 intensity mean ($N = 2$; $n_{RFP} = 17$, $n_{CRE} = 19$ cells). **E** Bar graph showing the somatic vGlut1 intensity mean ($N = 2$; $n_{RFP} = 17$, $n_{CRE} = 19$

cells). **F** Representative confocal (left column of panels) and STED images of autaptic hippocampal neurons labeled with nanobodies against vGlut1 (green) and PSD95 (magenta). Scale bars = 10 μm (left column of panels) and 5 μm (other panels). **G** Line plots showing the change in vGlut1 signal intensity against distance ($N = 3$; $n_{CTRL} = 13$, $n_{RFP} = 10$, $n_{CRE} = 13$ cells). **H** Line plots showing the change in PSD95 signal intensity against distance ($N = 3$; $n_{CTRL} = 13$, $n_{RFP} = 10$, $n_{CRE} = 13$ cells). AU arbitrary unit. In each graph, dots correspond to the total number of synaptic counts obtained from one individual neuron. Bars represent mean ± SEM. Data were compared using a Kruskal-Wallis, Dunn´s multiple comparison test, unpaired $t$ test, or Mann-Whitney test, where $*p < 0.05$, $**p < 0.01$.

showed that the vGlut1:vGlut2 ratio co-determines synaptic transmitter release probability, with vGlut2-expressing neurons exhibiting a higher release probability[45]. This observation could (partly) explain the release probability phenotype of Nedd8-KO neurons (Fig. 2). To test this possibility, we performed rescue experiments and either re-expressed vGlut1 or knocked-down vGlut2 (Supplementary Fig. 5A) in the Nedd8-KO background in order to restore a normal vGlut1:vGlut2 ratio, and studied the impact on the RRP and $P_{vr}$. However, neither overexpression of vGlut1 nor knockdown of vGlut2 rescued the RRP and $P_{vr}$ phenotype of Nedd8-KO neurons (Fig. 7A–F). Crucially, endophilin1 levels were also shown to correlate with the reduced release probability associated with vGlut1 expression in neurons[45]. Strikingly, Nedd8-KO causes a 50% reduction in endophilin1 expression that is detectable at both mRNA and protein levels (Fig. 7G–L). Moreover, neither re-expression of vGlut1 nor knockdown of vGlut2 rescued endophilin1 levels in Nedd8-KO cells (Fig. 7I–L). Thus, it is possible that restoration of vGlut1 or vGlut2 levels failed to rescue synaptic defects in Nedd8-KO neurons because of reduced endophilin1 level, which cannot be compensated by changing vGlut1:vGlut2 ratios.

### Neddylation regulates excitatory neurodevelopment
We show that Nedd8 ablation leads to minor morphological changes but strong alterations in presynaptic transmitter release probability and the

cycling of synaptic vesicles. To explore the biological processes underlying these morphological and functional defects, we examined changes in the neuronal transcriptional landscape of mature (DIV12) neurons upon Nedd8 deletion (Fig. 5). Our transcriptome analysis has revealed that Nedd8-KO alters the expression of genes encoding synaptic proteins involved in the release and cycling of synaptic vesicles, consistent with the functional deficits we observed. These phenotypes are likely to be the culmination of a more profound, as yet unknown, altered Nedd8-mediated signaling earlier in neuronal development. The most abundant Nedd8 targets are the cullins, which are large multiprotein cullin-RING E3 ubiquitin ligases[34,35] that regulate a variety of cellular processes, including neuronal development and dendritic arborization of neurons[36–38]. We therefore hypothesized that an impaired degradation of key regulatory factors was at the origin of the transcriptional and functional changes we observed in Nedd8-deficient neurons. For instance, our transcriptome analysis of matured neurons (DIV12) revealed a strong alteration in the expression profile of transcriptional regulators that are essential for the development and maturation of the glutamatergic neuron phenotype, e.g., *Pax6*, *Sox2*, *Neurogenin2*, *NeuroD1*, *NeuroD2*, or *Tbr1* (Supplementary Fig. 5B–G); changes that were further validated by RT-qPCR analyses (Supplementary Fig. 5H–M). Thus, it is likely that Nedd8-depletion alters, directly or indirectly, the protein expression levels of such regulators during

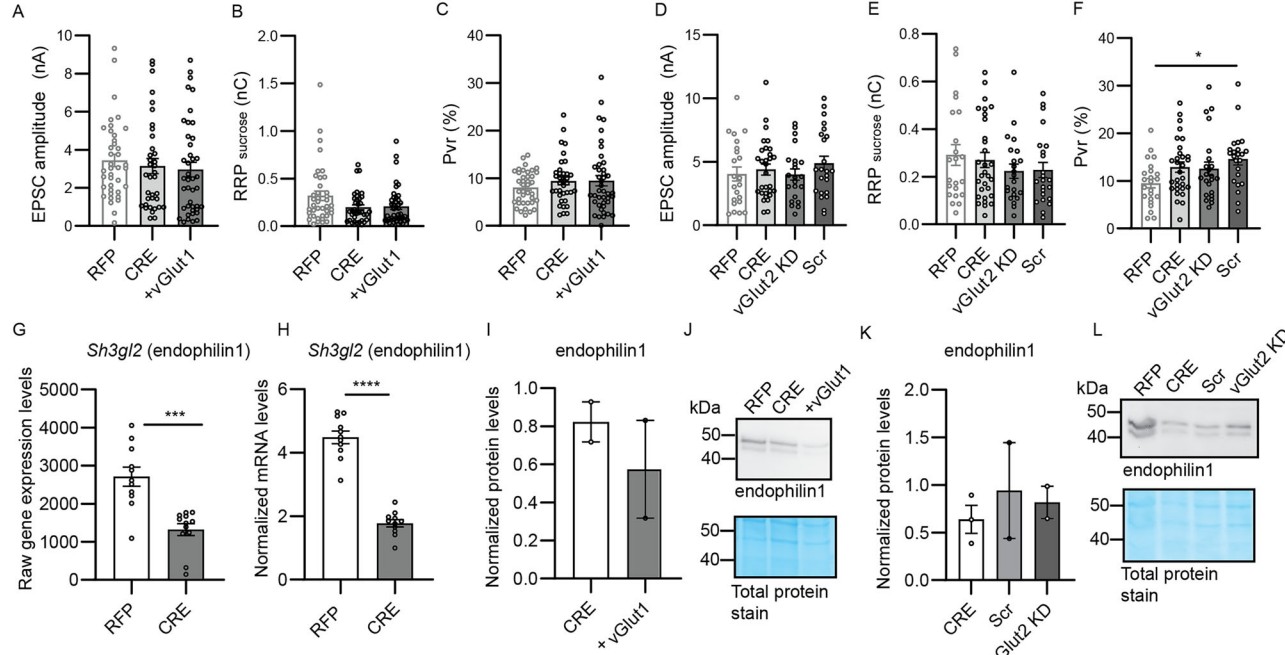

**Fig. 7 | Over expression of vGlut1, or knock-down of vGlut2 do not rescue the defects in release probability.** Hippocampal neurons from Nedd8cKO animals were infected at DIV1 with lentivirus expressing RFP, CRE-recombinase-RFP (CRE), co-infection with CRE and a virus expressing vGlut1 (+ vGLUT1), co-infection with CRE and a virus expressing shRNA to knock down the expression of vGlut2 (vGLUT2 KD), or co-infection with CRE and a virus expressing scrambled RNA sequences (Scr). Neurons were analyzed at DIV12. **A** Bar graph showing evoked EPSC amplitudes of autaptic excitatory neurons infected with lentivirus expressing RFP, CRE-recombinase-RFP (CRE), or CRE-recombinase-RFP (CRE) and a virus re-expressing vGlut1 (+ vGlut1) ($N = 2$, $n_{RFP} = 32$, $n_{CRE} = 26$, $n_{vGlut1} = 27$). Data was compared using a Dunn's multiple comparisons test, where $P_{RFP-CRE} = 0.91$, $P_{RFP-vGlut1} = 0.33$, $P_{CRE-vGlut1} > 0.99$. **B** Bar graph showing the total charge transferred by the release of the RRP upon application of a hypertonic sucrose solution in autaptic excitatory neurons infected with lentivirus expressing RFP, CRE-recombinase-RFP (CRE), or CRE-recombinase-RFP (CRE) and a virus re-rexpressing vGlut1 (+ vGlut1) ($N = 2$, $n_{RFP} = 32$, $n_{CRE} = 27$, $n_{vGlut1} = 28$). Data was compared using a Dunn's multiple comparisons test, where $P_{RFP-CRE} = 0.23$, $P_{RFP-vGlut1} = 0.13$, $P_{CRE-vGlut1} > 0.99$. **C** Bar graph showing the probability of vesicular release of autaptic excitatory neurons infected with lentivirus expressing RFP, CRE-recombinase-RFP (CRE), or CRE-recombinase-RFP (CRE) and a virus re-rexpressing vGlut1 (+ vGlut1) ($N = 2$, $n_{RFP} = 32$, $n_{CRE} = 27$, $n_{vGlut1} = 28$). Data was compared using a Dunn's multiple comparisons test, where $P_{RFP-CRE} > 0.99$, $P_{RFP-vGlut1} > 0.99$, $P_{CRE-vGlut1} > 0.99$. **D** Bar graph showing evoked EPSC amplitudes of autaptic excitatory neurons infected with lentivirus expressing RFP, CRE-recombinase-RFP (CRE), CRE-recombinase-RFP and a virus to knock down the expression of vGlut2 (vGlut2 KD) or with a control virus (Scr) ($N = 3$, $n_{RFP} = 15$, $n_{CRE} = 19$, $n_{vGlut2} = 16$, $n_{scramble} = 15$). Data was compared using a Tukey's multiple comparisons test, where $P_{RFP-CRE} = 0.95$, $P_{RFP-vGlut2} > 0.99$, $P_{RFP-Scr} = 0.65$, $P_{CRE-}$

$_{vGlut2} = 0.94$, $P_{CRE-Scr} = 0.88$, $P_{vGlut2-Scr} = 0.61$. **E** Bar grap of the total charge transferred by the release of the RRP upon application of a hypertonic sucrose solution in autaptic excitatory neurons infected with lentivirus expressing RFP, CRE-recombinase-RFP (CRE), CRE-recombinase-RFP and a virus to knock-down the expression of vGlut2 (vGlut2 KD) or with a control virus (Scr) ($N = 3$, $n_{RFP} = 15$, $n_{CRE} = 19$, $n_{vGlut2} = 16$, $n_{scramble} = 15$). Data was compared using a Dunn's multiple comparisons test, where $P_{RFP-CRE} > 0.99$, $P_{RFP-vGlut2} > 0.99$, $P_{RFP-Scr} > 0.99$, $P_{CRE-}$ $_{vGlut2} > 0.99$, $P_{CRE-Scr} > 0.99$, $P_{vGlut2-Scr} > 0.99$. **F** Bar graph showing the probability of vesicular release in autaptic excitatory neurons infected with lentivirus expressing RFP, CRE-recombinase-RFP (CRE), CRE-recombinase-RFP and a virus to knock down the expression of vGlut2 (vGlut2 KD) or with a control virus (Scr) ($N = 3$, $n_{RFP} = 15$, $n_{CRE} = 19$, $n_{vGlut2} = 16$, $n_{scramble} = 16$). Data was compared using a Dunn's multiple comparisons test, where $P_{RFP-CRE} = 0.29$, $P_{RFP-vGlut2} > 0.99$, $P_{RFP-}$ $_{Scr} = 0.014$, $P_{CRE-vGlut2} > 0.99$, $P_{CRE-Scr} > 0.99$, $P_{vGlut2-Scr} = 0.51$. **G** Bar graphs showing the raw gene expression levels of *Sh3gl2* (endophilin1, $N = 3$, $n_{RFP} = 11$, $n_{CRE} = 11$). Data was compared using a Mann-Whitney test, where $P = 0.0002$. **H** Normalized mRNA levels of *Sh3gl2* (endophilin1) as assessed by RT-qPCR ($N = 3$, $n_{RFP} = 11$, $n_{CRE} = 11$). Data was compared using an unpaired *t* test, where $P < 0.0001$. **I, J** Bar graph (**I**) showing normalized expression levels of endophilin1 as assessed by Western blot, as shown in (**J**) ($N = 2$, $n_{CRE} = 2$, $n_{vGlut1} = 2$). Molecular weight is indicated on the left side (kDa). **K, L** Bar graph (**K**) showing normalized expression levels of endophilin1 as assessed by Western blot as shown in (**L**) ($N = 2$, $n_{CRE} = 3$, $n_{scramble} = 2$, $n_{vGlut2} = 2$). Molecular weight is indicated on the left side (kDa). In graphs (**A–F**), dots correspond to measurements recorded from individual neurons. Bars represent mean ± SEM. Data were compared using a Kruskal-Wallis and Dunn´s multiple comparison test, unpaired *t* test, or Mann-Whitney test, where *$p < 0.05$, ***$p < 0.001$, ****$p < 0.0001$.

the course of neuronal maturation. Taken together, the failure to maintain adequate protein expression levels of key regulators of glutamatergic cell fate in Nedd8-deficient neurons likely explains the aberrant expression of synaptic genes required for the final expression of a mature glutamatergic identity and the aberrant synaptic transmission.

## Discussion

PTMs are essential throughout the lifetime of a nerve cell, from early development and differentiation to synaptic signaling and plasticity. The present study focused on neddylation, an enigmatic, Ubl-based PTM that has recently attracted major attention due to studies linking neddylation to neuronal development and synaptic transmission[28–33]. For optimal stringency in analyzing effects of perturbed neddylation, we employed a

conditional Nedd8-KO mouse line and studied the impact of Nedd8 deletion on neuronal development and synaptic transmission in hippocampal autaptic neurons. We found that Nedd8-deficient neurons (i) have a moderately reduced dendrite complexity but normal numbers of synapses (Fig. 1) and normal synapse ultrastructure (Supplementary Fig. 1), (ii) exhibit an increase in synaptic release probability, partly due to altered coupling between SVs in the RRP and VGCCs (Figs. 2 and 4), accompanied by a reduced paired-pulse ratio and more profound synaptic depression (Fig. 2H–L) and (iii) show no change in the RRP refiling rate normalized to the RRP$_{40Hz}$ size (Fig. 2N) or in VGCC function (Fig. 3).

On the other hand, Nedd8 deletion does not cause apparent postsynaptic defects. Some of these results, particularly the observation that postsynaptic features are unaffected by Nedd8-loss, are at odds with

previous findings[28,30,31]. This is likely due to differences in the neuron culture models used, the methodologies applied, and the developmental time point at which neddylation was perturbed. Whereas our approach exploited a genetic Nedd8-KO strategy to abolish neddylation during early neurodevelopment, other studies were either based on pharmacological perturbation of neddylation using the Nae1 inhibitor MLN-4924, which has known off-target effects[47,48], or on the ablation of Nae1 at a later developmental stage[28], thus precluding an assessment of the role of neddylation during earlier developmental timepoints.

Strikingly, our transcriptome analysis revealed that Nedd8-loss leads to a down-regulation of many genes encoding synaptic proteins. Most notably, we observed a strong decrease in vGlut1 expression, but a concomitant increase in vGlut2 expression upon Nedd8-deletion (Fig. 5 and Supplementary Data 2). This is consistent with previous studies linking reduced vGlut1 levels to neddylation defects[28,32,33], and resonates well with the correlation of synaptic vGlut1 and vGlut2 levels with respective lower and higher transmitter release probability[44,45]. While the total number of synapses was not changed in Nedd8-KO neurons, the 60–70% reduction in vGlut1 levels (Fig. 5) was paralleled by a substantial decrease in the number of vGlut1-containing synapses, without changes in the sub-synaptic distribution of the remaining vGlut1 (Fig. 6). The vGlut1-loss in Nedd8-KO neurons is offset by a 50% increase in vGlut2 levels (Fig. 5), which likely compensates for the loss of vGlut1, explaining why mEPSC amplitudes, which are co-determined by SV filling, are not affected by Nedd8-loss (Supplementary Figs. 1 and 2).

In view of the fact that high synaptic vGlut2 levels are linked to increased $P_{vr}$[44,45], we tested the hypothesis that the increased vGlut2-vGlut1 ratio might be at the basis of the increased $P_{vr}$ in Nedd8-KO neurons. This hypothesis was favored because VGCC function was unchanged in Nedd8-KO neurons (Fig. 3), and only slight changes in SV-VGCC coupling were observed (Fig. 4). However, neither re-expression of vGlut1 nor knock-down of vGlut2 in Nedd8-KO cells reverted the altered $P_{vr}$ (Fig. 7). We currently attribute this to reduced endophilin1 levels in Nedd8-KO neurons, which are not restored by vGlut1 re-expression or vGlut2 knock-down (Fig. 7), because the effect of vGlut1 expression levels on the $P_{vr}$ is known to be dependent on the presence of endophilin1[45,49]. In accord with this notion, the endocytotic regulator endophilin1 operates in release site clearance[50] and has been linked to RRP maintenance and the support of sustained exocytosis[51], which are compromised in Nedd8-KO neurons (Fig. 2).

To obtain deeper insight into the molecular mechanisms underlying deficits in Nedd8-KO neurons, we analyzed the neuronal transcriptome upon Nedd8 deletion. This approach was chosen to directly evaluate the impact of Nedd8-loss on the neuronal transcriptional landscape and to infer signaling events that can be linked to the physiological changes we have observed[52]. RNA-seq and differential expression analysis revealed an unexpected consequence of Nedd8 loss, characterized by a strong decrease in the expression of genes encoding synaptic proteins, especially when considering the overall upregulation of the rest of the transcriptome. After stringent filtering, 77% of all transcriptomic changes reflected an upregulation of gene expression upon Nedd8-loss (Fig. 5A). The subset of affected SynGO genes behaved very differently, with half upregulated and the other half downregulated, indicating that Nedd8-deletion strongly reduces the expression of genes encoding synaptic proteins, whereas the rest of the transcriptome is mainly upregulated.

Gene ontology and pathway analysis showed that upregulated genes represent a broad range of pathways that are not necessarily interconnected (Supplementary Fig. 4). In contrast, genes downregulated upon Nedd8-depletion represented a more restricted functional spectrum, with glutamatergic synapses, glutamatergic signaling, neuronal development, synaptic vesicle cycle, and synaptic plasticity featuring prominently (Supplementary Fig. 3). In particular, we found an imbalance in the vGlut1:vGlut2 expression ratio, which reflects the maturation of glutamatergic neurons. As neuronal morphology was only moderately affected while glutamatergic synaptic features showed strong changes upon Nedd8-loss, it appears that blocking neddylation impacts glutamatergic neurons quite specifically

rather than neurodevelopment in general. Overall, our data reveal a previously unknown role of neddylation in regulating the transcriptional landscape in neurons, with a strong influence on genes involved in excitatory synaptic signaling.

Given the substantial transcriptomic changes caused by Nedd8-depletion, the phenotypes we observed in Nedd8-KO neurons, including morphological changes and increased transmitter release probability, likely represent the final manifestations of a defined but complex Nedd8-dependent remodeling of the transcriptional landscape. This, in turn, is most likely linked to the function of cullins, which are the predominant Nedd8 substrates, part of the major class of cullin-RING E3 ubiquitin ligases, and involved in a broad range of biological processes, including cell growth, development, signal transduction, transcriptional control, and genome integrity[53,54].

As mentioned above, the differentiation of glutamatergic neurons entails a switch from vGlut2 to vGlut1 expression as the neurons mature[55]. In view of this, the increased vGlut1:vGlut2 ratio we observed in glutamatergic Nedd8-KO neurons might reflect defects in the differentiation and maturation program of these cells. Indeed, our transcriptome analysis revealed elevated levels of transcription factors found in neural progenitors (*Pax6*, *Sox2*, Supplementary Fig. 5), and decreased levels of markers for neural commitment (*NeuroD1*, *NeuroD2*, *NeuroG2*, *Trb1*, Supplementary Fig. 5)[11,12]. In general accord with the notion above, neddylation was reported to contribute to the regulation of neurogenesis and cell fate identity. For instance, Nae1 ablation in neuronal progenitor cells leads to severe morphological defects, likely due to impaired neurogenesis and neuronal differentiation[56]. Furthermore, Nae1 deletion in Schwann cells causes severe alterations in differentiation pathways, ultimately leading to peripheral neuropathies[57].

Altogether, our study shows that neddylation plays a key role in the differentiation and function of glutamatergic neurons. Our data indicate that Nedd8-ablation in young, post-mitotic, glutamatergic neurons causes alterations in the coordinated expression, activation, or homeostasis of developmental transcription factors that control neuronal differentiation, leading to aberrant transcriptional control and defects in the development of a mature glutamatergic neuronal phenotype. This becomes manifest in multiple ways, including increased vGlut2 expression level, reduced vGlut1 and endophilin1 expression levels, reduced dendrite complexity, and an increased probability of neurotransmitter release, partly due to altered coupling between SVs in the RRP and VGCCs.

## Methods
### Materials availability
The material described in this study is available upon request. This includes antibodies, DNA constructs or mouse lines.

### Animals
Generating *Nedd8* mouse mutant using CRISPR/Cas9 gene editing.

The Nedd8cKO knock-out mouse line was generated by site-directed CRISPR-Cas9 mutagenesis (LAVES Animals Protocol Permit 33.19-42502-04-19/3265). Superovulated FVB/N females were mated with FVB/N males, and fertilized eggs collected. In-house prepared CRISPR reagents (hCas9_mRNA, sgRNAs, preformed Cas9_sgRNA RNP complexes, and the dsDNA (HDR fragment) used as a repair template with the required two loxP insertions) were microinjected into the pronucleus and the cytoplasm of zygotes at the pronuclear stage using an Eppendorf Femtojet. Importantly, all nucleotide-based CRISPR-Cas9 reagents (sgRNAs and hCAS9_mRNA) were used as RNA molecules and were not plasmid-encoded, reducing the probability of off-target effects, due to the short live of RNA-based reagents[58,59]. The sgRNAs targeting the *Nedd8* Intron1 (201 bp upstream Exon2) and Intron3 (127 bp downstream Exon3) were selected using the guide RNA selection tool CRISPOR[60,61]. The correct site-specific insertion of the HDR fragment was confirmed by localization PCR with primers upstream and downstream of the HDR sequence, followed by cloning of the obtained PCR products and sequencing of the obtained

clones. The new line was designated Nedd8[em1Bros]. All oligonucleotide sequences are listed in Supplementary Data 1.

Mice were bred in the animal facility of the Max Planck Institute for Multidisciplinary Sciences (MPI-NAT, City Campus). Nedd8cKOs were classified as unburdened and showed no signs of pathogens, according to standard ethical guidelines. Animals were killed via cervical dislocation. We have complied with all relevant ethical regulations for animal use and received ethical approval from the Niedersächsisches Landesamt für Verbraucherschutz und Lebensmittelsicherheit (LAVES). Mouse maintenance and breeding were performed with permission of the LAVES. Animals were kept in ventilated cages (TYPE III, 800 cm2) at $21 \pm 1$ °C, 55% relative humidity and under a 12 h/12 h light dark cycle in specific pathogen-free conditions. Cages were changed once a week. Animals received *ad libitum* access to autoclaved food and water and were provided with bedding and nesting material. Animals' health was controlled daily by caretakers and a veterinarian. Health monitoring was done quarterly according to FELASA recommendations, with NMRI sentinels or animals from the respective colony. The sex of neonatal animals used for experiments was not checked.

### Genotyping strategy
For genotyping, genomic DNA (gDNA) was isolated from tail biopsies using a genomic DNA isolation kit (Nexttec, #10.924).

For the location PCR, 20 µL reactions were prepared using 1 µL clean gDNA (15–80 ng), 4 µL PrimerSet (4 pmol final each), 4 µL 5X Reaction Buffer (Finnzymes #F-524), 0.4 µL PhireHot-Start II Taq DNA Polymerase (Finnzymes #F-122L), 1 µL 10 mM dNTPs (Bioline #DM-515107), 4 µL Hi-Spec Additive (Bioline #HS-014101) and 5.6 µL H2O. Thermocycler parameters: 98 °C for 5 min, (98 °C for 45 s, 64 °C for 30 s, 72 °C for 60 s) repeated for 34 cycles, final step at 72 °C for 10 min.

For the diagnostic routine genotyping PCR, 20 µL reactions were prepared using 1 µL clean gDNA (15–80 ng), 4 µL PrimerSet (4 pmol final each), 4 µL 5X Reaction Buffer (Biozym #331620XL), 0.2 µL Hot-Start Taq DNA Polymerase (Biozym #331620XL), 1 µL 50 mM MgCl2 (AGCTLab stock) and 9.8 µL H2O. Thermocycler parameters: 96 °C for 3 min, (94 °C for 30 s, 62 °C for 60 s, 72 °C for 60 s) repeated for 32 cycles, final step at 72 °C for 7 min.

Primer sequence and fragment size information: Location PCR to confirm the site-specific insertion

Primer information: 38438 5'-CTAGGCCAAGTGTGGTGTTTGT GGT-3'; s_Up_HDR1_Nedd8-cKO location PCR; 38445 5'-AATCCCAGC AGACAAATCCATGTGA-3'; as_Down_HDR1_Nedd8-cKO location PCR

Fragment sizes: Nedd8_WT = 2983 bp (38438_38445_2983bp); Nedd8_cKO_floxed = 3051 bp (38438_38445_3051bp)

Routine genotyping diagnostic PCR:

Primer information: 38447 5'-CGTATTTAATTTGGAGCTTACC GTT-3'; Sense2_Nedd8_UpExon2; 38448 5'-GCAAAGATGGCGTAAG TCCCACA-3'; Asense3_Nedd8_UpExon2; 38450 5'-GGCACCTGTACT-CAAGCGTA-3'; Asense_Nedd8_DownExon3

Fragment sizes: Nedd8_UpEx2_WT = 302 bp (38447_38448_302bp; Nedd8_cKO_LoxP1 = 336 bp (38447_38448_336bp); Nedd8_KO = 388 bp (38447_38450_388bp), after Cre recombination

### Cell lines
HEK293-FT were kept in DMEM medium (Gibco) containing 10% FBS and antibiotics (Gibco, 100 µ/ml Penicillin, 100 µg/ml streptomycin, and 500 µg/ml gentamycin) at 37 °C with 5% CO2.

### Primary hippocampal neuron culture
Continental cultures of primary hippocampal neurons were performed as described[19]. Hippocampi were dissected from homozygous Nedd8cKO P0 mice. The tissue was digested in a solution containing 1 mM CaCl2, 0.5 mM EDTA, 1.65 mM l-cysteine in DMEM (Gibco) for 1 h at 37 °C under agitation (450 rpm). Digestion was stopped via a 15 min incubation at 37 °C in DMEM supplemented with 10% heat-inactivated fetal bovine serum (FBS, Gibco), 2.5 mg albumin, and 2.5 mg/ml trypsin inhibitor. The tissue was

washed twice with NBA medium (NBA, Gibco) containing, 2% B27 (Gibco), 2mM Glutamax (ThermoFisher), 100 U/ml of penicillin and 100 mg/ml of streptomycin (ThermoFisher). Careful trituration was performed manually in 100 µL of the same medium. Once the tissue was dissociated, the supernatant was plated onto 60 mm Petri dishes (CytoOne) previously coated with Poly-L-Lysine (Sigma) at an approximate density of 200.000 cells/well (two hippocampi). The medium was replaced the next day. Cultures were kept at 37 °C with 5% CO2.

### Primary autaptic culture
Primary autaptic neuronal cultures and astrocyte micro-island cultures were performed as previously described[62]. Briefly, hippocampi were dissected from homozygous Nedd8cKO P0 animals. Tissue digestion, washing and trituration steps were performed as described above for primary continental cultures. Neurons were plated on 6-well astrocyte micro-island plates at a density of 4.000 cells/well. Plates were kept in the NBA media containing Neurobasal A medium (Gibco), 2% B27 (Gibco), 2mM Glutamax (ThermoFisher), 100 U/ml of penicillin, at 37 °C and 5% CO2.

### Virus production
Lentivirus production was performed as previously described[63–66]. In brief, HEK293-FT (Thermo) cells grown in DMEM medium containing 10% FBS and antibiotics (100 µ/ml Penicillin, 100 µg/ml streptomycin, and 500 µg/ml gentamycin) with an 80–90% confluency were transfected with 20 µg of pVSVG packaging and pCMV delta R8.9 envelope constructs, and 40 µg of lentivirus construct expressing under the Synapsin1 promoter under the Synapsin1 promoter either red fluorescent protein (RFP) or CRE-recombinase tagged with an RFP protein (CRE). Transfection was performed using Lipofectamine 2000 (Thermo Scientific). After 6 h, the medium was changed to Optimem (Gibco) supplemented with 2% FBS, penicillin 100 µ/ml, Streptomycin 100 µ/ml, 2mM of Glutamax and 10 mM sodium butyrate. After 46–48 h, the cell supernatant was centrifuged at 2.000 rpm for 5 min at 4 °C and then filtered using a 0.45 µm filter (Merck). Viral particles were filtered and washed with Optimem once and twice with 1x tris-buffered saline (TBS) using a 100 kDa Amicon centrifugal filter (Millipore) via a series of centrifugation at 3.500 rpm for 10 min at 4 °C. Viral particles were concentrated to a final volume of 500 µL, and aliquots of 50 µL were snap frozen in liquid nitrogen. vGlut1 and vGlut2 rescue viral constructs, as well as empty control constructs, were provided by the Viral Core Facility of Charité Universitätsmedizin Berlin.

### Electrophysiology
Whole-cell patch-clamp recordings were performed in autaptic excitatory hippocampal neurons at DIV10-12. All recordings were done using a HEKA EPC-9USB amplifier controlled by PatchMaster software (Version 2×90.3, HEKA electronics), the recording rate was 10 kHz, and $R_{series}$ up to 10 MΩ with 50% compensation. The membrane potential of excitatory neurons was set to $-70$ mV in voltage-clamp configuration and depolarized to 0 mV to induce action potentials. For synaptic properties, the extracellular solution contained (mM); 140 NaCl, 2.4 KCl, 10 HEPES, 10 Glucose, 4 CaCl2, 4 MgCl2, pH 7.6. Microelectrodes were fabricated from borosilicate glass pipettes using a micropipette puller (P-27, Sutter Instrument) and filled with intracellular solution containing (mM) 136 KCl, 17.8 HEPES, 1 EGTA, 4.6 MgCl2, 4 NaATP, 0.3 Na2GTP, 15 creatine phosphate, and 5 µ/ml phosphocreatine kinase osmolarity 315–320 mOsmol/L, pH 7.4. Pipette resistances were around ~ 2–4 MΩ. Excitatory post-synaptic currents (EPSC) were recorded using frequencies of 0.2, 10 and 40 Hz. The release of vesicles from the readily releasable pool (RRP) was triggered by application of a hypertonic 0.5 M sucrose solution. Estimation of the size of the RRP was obtained from the hypertonic sucrose treatment and back extrapolation of the cumulative EPSC from 40 Hz trains of action potential[67]. Miniature EPSC (mEPSC) were recorded under 300 mM tetrodotoxin (TTX, Tocris) treatment, and for post-synaptic responses, 100 µM Glutamate (Sigma) or 3 µM γ-aminobutyric acid (GABA) (Sigma) solutions were used. The paired-pulse ratio was calculated from the 40 Hz train stimulus and is the

ratio of the peak amplitudes of the second and first responses. For calcium channel sensitivity studies, the extracellular solution contained (mM) 140 TEA-Cl, 2.5 CsCl, 10 CaCl$_2$, 1 MgCl$_2$, 10 HEPES, 10 glucose, Osmolarity 310 mOsm, pH 7.3; and the intracellular solution contained 130 CsCl, 10 HEPES, 2 CaCl$_2$, 10 EGTA, 5 MgATP, Ph 7.3. 10 µM Nimodipine (Alomone Labs) or 0.2 µM ω-Agatoxin (Alomone Labs) (L-type and P/Q-type calcium channel blocker, respectively) were used to block specific calcium channels. For synaptic vesicle coupling studies, 100 µM EGTA-AM (Invitrogen) was used. Infected neurons were distinguished by a nuclear RFP signal that was visualized using an inverted microscope (Zeiss). Data were analyzed using AxoGraph software (Version 1.5.4., Axograph Scientific).

### Western blot

Neuronal cultured were lysed in lysis buffer containing 20 mM Tris Base, 150 mM NaCl, 1% Triton X-100 (Roche), and protease inhibitors (0.5 µg/ml aprotinine, 0.2 mM phenylmethylsulfonyl fluoride, 1 µg/ml leupeptine, 20 mM N-ethylmaleimide), pH 7.4–7.6. For storage, samples were snap-frozen in liquid nitrogen and kept at − 80 °C. SDS-PAGE was performed using self-made 10% acrylamide gels, or commercial, pre-casted 4–12% Bis-Tris gels (Thermo Scientific). Samples were sonicated, and protein concentration was determined using the Bradford assay (Biorad). Samples were heated for 5 min at 95 °C, or for 20 min 65 °C, shortly before loading. Gel migration was performed in a buffer containing 5 mM MOPS, 5 mM Tris Base, 0.01% SDS, 0.1 mM EDTA, pH 7.7. Proteins were transferred to nitrocellulose membranes (Cytiva) using a buffer containing 192 mM Glycine, 25 mM Tris Base, and 20% Methanol. Total protein content was visualized using the Memcode reversible protein staining kit (Thermo Scientific). Primary and secondary antibody incubation of the membrane were performed in 1x PBS supplemented with 5% milk (Frema) and 0.1% Tween 20 (PBS-T) following standard procedure. Signal was revealed using enhanced chemiluminescent (GE Healthcare), and signals were detected with an ECL Chemostar Imager (Intas Science Imaging) and analyzed with Fiji (NIH). Quantification was performed by normalizing the signal in each lane with its respective Memcode signal.

### RNA isolation, RNA-sequencing, and analysis

The continental culture of primary hippocampal neurons aged for 13 days were subjected to TRIzol-RNA extraction following the manufacture protocol (Zymo). RNA quality was assessed by measuring the RNA integrity number (RIN) using a Fragment Analyzer HS Total RNA Kit (Advanced Analytical Technologies, Inc.). Library preparation for RNA-Seq was performed in the STAR Hamilton NGS automation using the Illumina Stranded mRNA Prep, Ligation (Cat. N°20040534 and the Illumina RNA UD Indexes Set A, Ligation, 96 Indexes, 96 Samples Cat. N°20091655), starting from 300 ng of total RNA. The size range of the final cDNA libraries was determined by applying the SS NGS Fragment 1- to 6000-bp Kit on the Fragment Analyzer (average 340 bp). Accurate quantification of cDNA libraries was performed by using the QuantiFluor™ dsDNA System (Promega). cDNA libraries were amplified and sequenced by using an S4 flow cell NovaSeq6000; 300 cycles, 25 Mio reads/sample from Illumina.

**Raw read & quality check.** Sequence images were transformed with BaseCaller Illumine Software to BCL files and demultiplexed to fast files with bcl2fastq v2.20.0.422. The sequencing quality was asserted using FastQC v.0.11.5 software (http://www.bioinformatics.babraham.ac.uk/projects/fastqc/)[68].

**Mapping and normalization.** Sequences were aligned to the reference genome Mus musculus (GRCm39 version 110, https://www.ensembl.org/Mus_musculus/Info/Index) using the RNA-Seq alignment tool (version 2.7.8)[69], allowing for 2 mismatches within 50 bases. Subsequently, read counting was performed using featureCounts[70]. Read counts were normalized in the R/Bioconductor environment (version 4.3.1) using the DESeq2[71] package version 1.40.2.

**PCA and other quality checks.** Principal component analysis (PCA) was performed using the sklearn Python library (version 1.3.1) (Supplementary Fig. 2B. 1). The total gene count box plot displays gene expression levels in each of the samples analyzed Supplementary Fig. 2B. The horizontal line inside each box represents the median gene expression. The box itself encompasses the interquartile range, covering from the 25th to the 75th percentile. The whiskers extend to the 10th percentile on the lower end and the 90th percentile on the higher end. Any points outside the whiskers, particularly those above the 90th percentile, are shown as individual circles. The plot uses color to differentiate between conditions, with red indicating the 'CRE' condition and green representing the 'RFP' condition. After PCA and gene count analysis, two samples were removed from the differential analysis: Nedd8Cre1-3 and Nedd8Cre3-3.

**Differential expression and pathway enrichment analysis.** Differential expression analysis was performed using PyDESeq2, a Python implementation of DESeq2 method[71,72] (version 0.4.4). Pathway enrichment was done with GSEA method[73] using blitzgsea library (version 1.3.42)[74]. The calculation of the score in pathway analysis was done as previously described[75].

### Real time quantitative PCR (RT-qPCR)

Continental cultures of DIV13 primary hippocampal neurons were subjected to TRIzol-RNA extraction according to the manufacturer's protocol (Zymo). RNA extraction and purification were performed using the Direct-zol RNA microprep Kit (Zymo) according to the manufacturer's instructions. Reverse transcription was performed with 500 ng of total RNA using SuperScript III Reverse Transcriptase (Invitrogen) according to the manufacturer's instructions. qPCR was performed with a Mic qPCR Cycler (Bio Molecular Systems). All genes were analyzed using Power SYBR™ Green PCR Master Mix (Applied Biosystems). For each reaction, 1 µL of cDNA, 2 pmol/µL of primers, and 16 µL of SYBR Green Master Mix were used. All genes were tested in duplicate. For the analysis, all the Ct values were standardized to the Ct of *Gapdh* and *Ubc*, and normalized to the levels of the RFP condition. The list of primers used for RT-qPCR is available in Supplementary Data 1.

### Immunostaining and confocal microscopy

DIV12-13 autaptic hippocampal neurons were fixed with 4% paraformaldehyde (PFA) prepared in 1x phosphate-buffered saline (PBS) for 10 min. All steps were done at room temperature. Neurons were washed three times with 1x PBS and quenched for 10 min with 50 mM Glycine in 1x PBS. Neurons were permeabilized in 1 x PBS containing 0.1% Triton X-100 (Roche), 2.5% goat serum (GS) (Gibco) for 30 min. Cells were washed in 1 x PBS supplemented with 2.5% GS and incubated with primary antibodies diluted in the same solution for 1 h. Neurons were washed and incubated with secondary antibodies for 45 min in darkness. Coverslips were washed in 1x PBS and mounted on slides using Aqua-Poly Mount (Polysciences) medium and stored at 4 °C until image acquisition. Images were obtained with a SP8 confocal Leica microscope (Leica Microsystems), 63x immersion objective (NA 1.4), 12-bit image depth, step size of 0.3 µm, and resolution of 1232 × 1232 corresponding to a pixel size of 149.9 nm. Synapse counting, Sholl analysis and total dendrite length were calculated using Imaris software (Version 9.9.1, Oxford Instruments). Representative images were processed using Fiji (Version 2.14.0/1.54 f, NIH).

### STED microscopy and analysis

DIV12 autaptic hippocampal neurons were fixed with 4% PFA for 45 min and then quenched using 100 mM NH$_4$Cl in 1 x PBS for 20 min. All steps were done at room temperature. Cells were washed twice in 1 x PBS and incubated in a blocking solution containing 2.5% bovine serum albumin (BSA, Biomol) and 0.1% Triton X-100 (Roche) in 1 x PBS. Neurons were incubated with nanobodies against vGlut1-AZD568 (Nanotag),

PSD95-STAR635 (Nanotag), and Synaptotagmin1-ATTO488 (Nanotag) in a 1:100 dilution in blocking solution. Coverslips were washed twice in 1 x PBS and mounted on slides using Mowiol (Carl Roth). Image acquisition was done using an Abberior Expert line setup (Abberior Instruments) with an IX83 microscope (Olympus), with a 100x immersion objective (1.4 NA). Excitation of STAR635 was performed with a 640 nm laser (5% of max. power) and detected with an avalanche photodiode (APD) laser (range 650–720 nm). For AZD568, excitation was done using a 580 nm laser (30% of max. power) and detected with an APD laser (range 605–625 nm). For both STAR635 and AZD568, depletion was achieved with a 775 nm depletion laser (20% and 60%, respectively). Excitation of ATTO488 was achieved with a 488 nm laser (4% of max. power) and detected with an APD laser (range 525–575 nm). For ATTO488, a solid-state 595 nm depletion laser (10% of max. power) was used. For each neuron, 2–5 regions of interest were selected and imaged using confocal and STED microscopy. Pixel sizes were 200 and 20 nm for confocal and STED, respectively, dwell times 8–10 μs, and lines accumulation of 3–5 for STED. The analysis was performed using MATLAB (Version R2022b, The Mathworks, Inc.). Synapses were identified automatically, using a band-pass filter procedure, which revealed the synapse-sized spots in the different channels. Spots of less than 9 pixels were removed, being considered to be noise events. All synapse images from one experiment were then automatically rotated and oriented to display the postsynaptic density in the center, and the presynapse to the left. Line scans were generated across the synapses, and the intensities in the different channels were measured and displayed. Representative images were processed using Fiji (Version 2.14.0/1.54 f, NIH).

### Ultrastructural analysis of Nedd8-deficient synapses
**Primary hippocampal neuron culture.** For electron microscopy experiments, neuronal monolayer cultures from P0 Nedd8-cKO mice were grown directly on carbon- and PDL-coated 6 mm sapphire disc freezing substrates (Wohlwend). Sapphire discs were mounted on 18 mm glass coverslips using Matrigel and placed in a 12-well plate for UV sterilization for 1 h at RT. Hippocampi were prepared as described above in this section. Cultured neurons were vitrified by high-pressure freezing and processed for ultrastructural analysis at DIV13.

**High pressure freezing, freeze-substitution, and ultramicrotomy.** Sapphire discs with neurons were infected at DIV1 with lentiviral constructs designed to express Cre-recombinase and RFP, or RFP alone (negative control). At DIV 13, infected neurons underwent three washing steps in pre-warmed (37 °C) conditioned medium before rapid cryo-fixation using an EM ICE high-pressure freezer (Leica) and subsequent storage in liquid nitrogen. Semi-automated freeze-substitution was performed using an AFS2 device (Leica) equipped with customized aluminum sapphire disc revolvers, and samples were subsequently embedded in epoxy resin according to previously published protocols[76,77]. Sapphire discs were separated from polymerized samples with a razor blade, and resin block-faces were trimmed with an EM TRIM high-speed milling device (Leica) in preparation for ultramicrotomy. A 35° Ultra-Semi diamond knife (Diatome) mounted on a UC7 ultramicrotome (Leica) was used to cut 250 nm-thick sections onto formvar-coated copper parallel line grids (Gilder; G100P-Cu). Mounted sections were coated on both surfaces with Protein A-conjugated 10 nm gold fiducial particles (Cell Microscopy Core Products, University Medical Center Utrecht, The Netherlands) in preparation for 3D ultrastructural analysis using transmission electron tomography.

**Electron microscopy.** Synapses were identified and selected for 3D ultrastructural analysis in tiled overviews acquired at 5000 x magnification using a 200 kV Talos F200C G2 scanning/transmission electron microscope equipped with a 16 MP Ceta CMOS camera (Thermo Scientific) and SerialEM software (*IMOD* Software Package[78]). Tilt series ( ± 60° tilt range; 1° tilt increments) were acquired at 36000 x (unbinned pixel size = 0.4 nm) magnification from orthogonal axes using a Model 2040 dual-axis specimen holder (Fischione). Tomograms were generated from tilt-series using Gaussian filtering and a weighted back-projection algorithm implemented using *Etomo* software (IMOD software package[79]). Presynaptic vesicles were initially segmented using SynapseNet[80] as perfect spheres and then proof-read using 3dmod software (*IMOD Software PackMastronadeage*[79]). Active zones were segmented manually on every 3$^{rd}$ tomographic slice. Segmented data were analyzed using *mtk* and *imodinfo* programs in conjunction with a customized Python script to quantify the relative spatial distribution of presynaptic vesicles (i.e., closest approach to active zone release sites), their size (i.e., SV diameter) and clustering characteristics (i.e., inter-vesicular nearest-neighbor distances), respectively.

**Statistical analysis and reproducibility.** All statistical analyses were performed using GraphPad Prism software (Version 10.3.0., GraphPad). Normal distribution was determined using D´Agostino-Pearson or Shapiro-Wilk normality tests. Statistical analysis was performed using a Kruskal-Wallis and Dunn´s multiple comparison test, unpaired $t$ test, Welch's test, or Mann-Whitney tests. For graphs, data are shown as mean ± SEM. For all figures, statistically significant differences are denoted in graphs with * standing for $p$-value, where * refers to $p < 0.05$, ** to $p < 0.01$, *** to $p < 0.001$, and **** to $p < 0.0001$. N refers to the number of animals or biological replicates, and n corresponds to the number of cells. N and n are indicated in the legends.

### Reporting summary
Further information on research design is available in the Nature Portfolio Reporting Summary linked to this article.

### Data availability
RNA-seq data are accessible under the GSE269898 and are publicly available as of the date of publication. All numerical data are available in Supplementary data files 2–7. Supplementary Data 1 entails the list of Key resources reagents (antibodies, oligonucleotides). Supplementary Data 2 entails the list of differentially expressed genes upon KO of Nedd8. Supplementary Data 3 entails all the numerical data for the Sholl analysis (Fig. 1D–G), confocal and STED imaging (Figs. 1H, 6A, F). Supplementary Data 4 entails all the numerical data for electrophysiological recordings (Figs. 2, 3, 4, 7A–F). Supplementary Data 5 entails all numerical data regarding the analysis of the RNA-seq (Figs. 5B, C, 7H and Supplementary Fig. 2D, 5B–G), Western blot (Figs. 1B, 5F, G, 7I, K) and qPCR experiments (Figs. 5D, E, 7H and Supplementary Fig. 2E, 5H–M). Supplementary Data 6 entails all numerical data for Supplementary Fig. 1. Supplementary Data 7 entails all numerical data for supplementary Figs. 2, 3 and 4. All original, uncropped blots can be found in Supplementary Fig. 6. All data and supplementary data described in this study will be shared upon reasonable request to the lead contact. Any additional information required to reanalyze the data reported in this work is available from the lead contact upon request.

### Code availability
This paper does not report original code. Jupiter Notebook for graph generation can be shared upon request.

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

## Acknowledgements

The authors have no conflicting financial interests. We thank D. Warnecke, and C. Harenberg for their expert technical assistance in DNA synthesis, DNA sequencing, and mouse genotyping, the staff of the Max Planck Institute for Multidisciplinary Sciences Transgenic Animal Facility for the generation and maintenance of mouse colonies, Tanja Nilsson, Sally Wenger, and Sabine Beuermann for their technical support. RNA-sequencing was performed at the NGS Integrative Genomics (NIG) Core unit of the Institute of Pathology at the University Medical Center Goettingen. We thank Thorsten Trimbuch and Bettina Brokowski from the viral core facility of the Charité – Universitätsmedizin Berlin for providing vGlut1 and vGlut2 rescue constructs. This work was supported by the German Research Foundation (SFB1286/A1/A3/A9, J.T., Z.V., M.F., S.B., M.T., N.B., B.H.C., S.O.R.).

## Author contributions

M.T. conceived the project, performed experiments, and wrote the manuscript with the help of all other co-authors. J.T. and Z.V. contributed to the project plan, performed experiments, and wrote the manuscript. B.H.C. prepared samples for electron microscopic analysis and acquired tomograms. V.S. and I.H.G.P. analyzed the E.M. dataset. M.F. and S.B. analyzed the RNA-seq dataset. F.B. designed the CRISPR/Cas9-based Nedd8-cKO strategy and validated the CRISPR material, genotyped the offspring, and designed primers for RT-qPCR analysis. S.O.R. analyzed the STED microscopy images and provided material for imaging. J.S.R. supervised the electrophysiological experiments. N.B., J.S.R., B.H.C., F.B., and S.O.R. provided material, conceptual feedback, and edited the manuscript.

## Funding

## Competing interests

The authors declare no competing interests.
