## [Transparent Peer Review file · Communications Biology]

Neddylation regulates the development and function of glutamatergic neurons

Corresponding Author: Dr Marilyn Tirard

Version 0:

Reviewer comments:

Reviewer #1

(Remarks to the Author)

In this article, the authors generated a new model of neddylation inhibition (Nedd8 cKO mice) and analysed its impact of Nedd8 loss on neuronal development and synaptic transmission. They observed defects in neuronal morphology and synaptic vesicle (SV) release probability, due to altered levels of the vesicular glutamate transporters vGlut1 and vGlut2 and of the endocytosis regulator endophilin1. Their data also point to alterations in the expression of key synapse proteins but also of transcriptional regulators involved in neuronal development and differentiation.

The authors used a variety of techniques, and their results are broadly in line with their conclusions. These are some points to address:

1) In the discussion, the authors refer to the discrepancy in effects of neddylation inhibition on postsynaptic features of their study with previous findings (Vogl et al., 2015, Brockmann et al., 2019, Scudder and Patrick, 2015), as a result of the different neuron culture models used, and the different methodology employed to perturb neddylation. For instance, the authors used a genetic Nedd8-KO strategy to abolish neddylation while they mention that the other studies were based on the pharmacological perturbation of neddylation using the Nae1 inhibitor MLN-4924, which has known off-target effects (Mao and Sun, 2020, Zhang et al., 2022).

However, this is not exactly true, since in the study by Vogl et al., 2015, they also performed genetic inactivation of neddylation (by ablating the Nae1 enzyme) and found strong effects even in vivo. The authors should explain better these discrepancies.

2) Please explain similar discrepancies in synapse numbers between this study and Vogl et al, 2015 study?

3) It is unclear if the effects on Pax6, Ngn2 and Sox2 levels are statistically significant (Figure 7 N-P)?

They mention "While Pax6 levels decreased during WT culture maturation, Pax6 levels remained stable in Nedd8-deficient neurons (Figure 7N). Regarding Ngn2, its levels increased during culture maturation, but not in Nedd8-depleted neurons (Figure 7O). Finally, Sox2 levels remained low in WT conditions, whereas Sox2 levels increased during culture maturation in Nedd8-KO neurons (Figure 7P)."

Statistical significance should be shown to back up these claims, and if not, the authors should redraw their conclusions.

4) It would be more appropriate to perform a proteomics analysis to truly identify the key regulators, since neddylation is a PTM of proteins, and direct targets of neddylation could be identified. It would also be important to determine how neddylation affects their targets (by examining protein stability, etc.) or if it involves degradation by CRLs.

Reviewer #2

(Remarks to the Author)

This manuscript investigates the role of Nedd8 in the development and function of excitatory neurons, focusing on transcriptional regulation, electrophysiology, and neuronal morphology using a conditional knock-out (cKO) mouse model

and derived cell cultures. This manuscript addresses an important topic, but some aspects could be improved.

- 1) I am not impressed by the effect sizes presented in some ephys results (e.g. Figs 2B, C, J,K). How can that implicate functional relevance?
- 2) Some blots on Fig 7A,N,O,P appeared too faint. Please show clearer and stronger bands.
- 3) Introduction and discussion could be revised to become more to the point and less of a review article.
- 4) Though the transcriptome analysis is interesting, obtaining individual gene expression from the bulk data is not sufficient to support claims of impaired neuronal development and endophilin expression. This should be confirmed in additional culture experiments.

Version 1:

Reviewer comments:

Reviewer #1

(Remarks to the Author)

The authors have adequately answered this reviewer comments.

Reviewer #2

(Remarks to the Author)

The authors have addressed all my concerns.

Point-by-Point Response (in Blue) to Reviewers' Comments

Reviewer 1

In this article, the authors generated a new model of neddylation inhibition (Nedd8 cKO mice) and analysed its impact of Nedd8 loss on neuronal development and synaptic transmission. They observed defects in neuronal morphology and synaptic vesicle (SV) release probability, due to altered levels of the vesicular glutamate transporters vGlut1 and vGlut2 and of the endocytosis regulator endophilin1. Their data also point to alterations in the expression of key synapse proteins but also of transcriptional regulators involved in neuronal development and differentiation.

The authors used a variety of techniques, and their results are broadly in line with their conclusions. These are some points to address:

1) In the discussion, the authors refer to the discrepancy in effects of neddylation inhibition on postsynaptic features of their study with previous findings (Vogl et al., 2015, Brockmann et al., 2019, Scudder and Patrick, 2015), as a result of the different neuron culture models used, and the different methodology employed to perturb neddylation. For instance, the authors used a genetic Nedd8-KO strategy to abolish neddylation while they mention that the other studies were based on the pharmacological perturbation of neddylation using the Nae1 inhibitor MLN-4924, which has known off-target effects (Mao and Sun, 2020, Zhang et al., 2022).

However, this is not exactly true, since in the study by Vogl et al., 2015, they also performed genetic inactivation of neddylation (by ablating the Nae1 enzyme) and found strong effects even in vivo. The authors should explain better these discrepancies.

2) Please explain similar discrepancies in synapse numbers between this study and Vogl et al, 2015 study?

It is possible that the discrepancies concerning the effects of blocked neddylation on postsynaptic features and on total synapse number can be attributed to methodological differences between our study and that of Vogl et al. (2015). First, Vogl et al. (2015) mostly used primary neuron cultures from rat brains, whereas we used mouse primary neurons. While the two rodent models share many similarities, it is conceivable that aspects of the neddylation dynamics and neddylation targets differ between the two models. A similar observation was made in the context of sumoylation (Daniel et al., 2018; PMID 28598330). Second, and most importantly, the developmental timepoint at which Nedd8 was ablated differs between the study by Vogl et al. (2015) and our study. For instance, Vogl et al. (2015) transfected neurons later (DIV 12 for primary neurons; DIV17 when using the inducible Ubc12-C111S, a negative dominant form of the Nedd8 E2), while we deleted Nedd8 at DIV1. Third, we ablated Nedd8 via a conditional KO approach. This leads to an almost complete loss of Nedd8 expression (our Figure 1 and Supplementary Figure 2D and 2E), whereas Nedd8 levels were not depleted to the same extent in the study by Vogl et al. (2015). Regarding in vivo data, Vogl et al. (2015) used Nae-CamKIIalpha-CreERT2, thereby achieving Nedd8 KD by tamoxifen injection starting after birth (P35 or postnatal week 7/8), and the spine analysis was performed at a much later stage (P50 or postnatal week 11/12). Here again, the timing of neddylation inhibition differs from our study. Altogether, the technical and methodological differences between our study and the findings described by Vogl et al. (2015) likely account for the different results, lending major relevance to both studies. We added a corresponding passage to the discussion part of our revised manuscript (p. 8/9).

3) It is unclear if the effects on Pax6, Ngn2 and Sox2 levels are statistically significant (Figure 7 N-P)? They mention "While Pax6 levels decreased during WT culture maturation, Pax6 levels remained stable in Nedd8-deficient neurons (Figure 7N). Regarding Ngn2, its levels increased during culture maturation, but not in Nedd8-depleted neurons (Figure 7O). Finally, Sox2 levels remained low in WT conditions, whereas Sox2 levels increased during culture maturation in Nedd8-KO neurons (Figure 7P)."

Statistical significance should be shown to back up these claims, and if not, the authors should redraw their conclusions.

We are grateful for this comment. We have now revised our conclusions. Despite numerous trials, and mainly because of the weak specificity of the antibodies used, we remain unable to obtain statistical significance regarding the changes in the protein levels of the selected transcriptional regulators of neuronal development. Therefore, we removed the corresponding dataset change the text of the revised manuscript accordingly.

4) It would be more appropriate to perform a proteomics analysis to truly identify the key regulators, since neddylation is a PTM of proteins, and direct targets of neddylation could be identified. It would also be important to determine how neddylation affects their targets (by examining protein stability, etc.) or if it involves degradation by CRLs.

We thank the reviewer for this very pertinent comment. Indeed, it would be ideal to identify neuronal Nedd8 targets and to determine how their neddylation is linked to the neurodevelopmental and functional phenotypes we observed. We are currently actively working on such a strategy. For instance, we are in the process of generating an HA-Nedd8 knock-in mouse model, analogous to the ones we developed for SUMOs (PMIDs 37009224, 2963341, 28598330, 23213215). In a parallel approach, we are developing anti-Nedd8 affinity purification protocols for mouse tissue and primary cells. As these approaches, as well as the subsequent mass-spectrometry analyses and follow-up, are time-consuming and very labor-intensive, we believe that such analyses extend beyond the scope of the present submission. In essence, we cannot provide Nedd8 targets at this juncture and ask sincerely that the requirement for obtain such data for a revision be waived.

Reviewer 2

This manuscript investigates the role of Nedd8 in the development and function of excitatory neurons, focusing on transcriptional regulation, electrophysiology, and neuronal morphology using a conditional knock-out (cKO) mouse model and derived cell cultures. This manuscript addresses an important topic, but some aspects could be improved.

1) I am not impressed by the effect sizes presented in some ephys results (e.g. Figs 2B, C, J,K). How can that implicate functional relevance?

We acknowledge the point raised by this reviewer but ask respectfully to consider the following. First, our data were acquired in five different biological replicates and involved 30-42 cells per dataset in total - more than is often done by other labs. We therefore feel that any changes we see are - in principle - biologically meaningful. Second, it is important to note that even small changes in RRP size (Figure 2B) and particularly in synaptic release probability can have substantial effects on circuit function. Along these lines, Figures 2H, 2J, and 2K show that the altered synaptic release probability we observe in Nedd8-deficient neurons leads to changes in synaptic short-term plasticity - here synaptic depression. Such changes in synaptic short-term plasticity are well known in the field to affect circuit logic and function (e.g. Deng et al., 2024, PMID 39303044; Asopa and Bhalla, 2023, PMID 37245258; Regehr, 2012, PMID 22751149).

We did not further emphasize this point in the revised manuscript, but would be willing to if required. We trust that the data speak for themselves.

2) Some blots on Fig 7A,N,O,P appeared too faint. Please show clearer and stronger bands.

Please see our response to comment 4 of Reviewer 1. The corresponding data have been removed from the revised manuscript.

3) Introduction and discussion could be revised to become more to the point and less of a review article.

We made a serious effort to further improve the indicated manuscript parts, also in response to Reviewer 1. We hope our introduction and discussion are now adequate.

4) Though the transcriptome analysis is interesting, obtaining individual gene expression from the bulk data is not sufficient to support claims of impaired neuronal development and endophilin expression. This should be confirmed in additional culture experiments.

Following this reviewer's comment, we pursued two approaches:

1. To further support our hypothesis regarding the role of Nedd8 in regulating neuronal development, we performed qPCR using the original RNA used for the RNA-seq analysis ($n_{RFP}=11$, $n_{CRE}=11$). This analysis (Figure 5D, 5E, 7H; Supplementary Figure 2D and 2E, Supplementary Figure 5H-M) confirms and further strengthens our interpretation of our data.

2. We performed qPCR using a second, independent set of samples ($n_{CRE}=3$, $n_{RFP}=3$), and obtained similar results as the ones observed originally with the samples sent for RNA-seq (see image below). For the sake of clarity and to not overload the manuscript, which contains seven main and five supplementary figures, and because the second RNA dataset repeats the first ones, we decided to not include the new dataset in the revised submission. However, we are willing to do this if required.

Figure 1: Validating repetition of RT-qPCR analysis. A-J. Bar graphs showing normalized mRNA levels of genes as indicated on top of each graph. N=3 for each group. Bar represent mean \pm SEM. Data was compared using a Mann-Whitney test, where * $p < 0.05$.

Beyond the changes outlined above, we decided to include a dataset on the ultrastructural organization of *Nedd8*-deficient synapses (Supplementary Figure 1). The corresponding data show that there are no major ultrastructural changes in *Nedd8*-deficient synapses, indicating that the activity-dependent changes in synaptic release probability we observed originate from deeper molecular *Nedd8*-dependent mechanisms, as revealed by our RNA-seq approach. Despite the lack of significant differences within this dataset, we believe that these data are important, to justifying their inclusion in the revised submission.